# FIG: Flow with Interpolant Guidance for Linear Inverse Problems

**Yici Yan[2*], Yichi Zhang[1*], Xiangming Meng[3], Zhizhen Zhao[1]**

[1]Department of Electrical and Computer Engineering, University of Illinois at Urbana-Champaign
[2]Department of Statistics, University of Illinois at Urbana-Champaign
[3]ZJU-UIUC Institute, Zhejiang University

## Abstract

Diffusion and flow matching models have recently been used to solve various linear inverse problems in image restoration, such as super-resolution and inpainting. Using a pre-trained diffusion or flow-matching model as a prior, most existing methods modify the reverse-time sampling process by incorporating the likelihood information from the measurement. However, they struggle in challenging scenarios, such as high measurement noise or severe ill-posedness. In this paper, we propose Flow with Interpolant Guidance (FIG), an algorithm where reverse-time sampling is efficiently guided with measurement interpolants through theoretically justified schemes. Experimentally, we demonstrate that FIG efficiently produces highly competitive results on a variety of linear image reconstruction tasks on natural image datasets, especially for challenging tasks. Our code is available at: https://riccizz.github.io/FIG/.

## 1 Introduction

Linear inverse problems have long been an active field of research in applied mathematics, statistics, and signal processing. In particular, linear inverse problems in image restoration have many real-world applications. A typical mathematical model for linear inverse problems reads as follows:

$$\boldsymbol{y} = \boldsymbol{A}\boldsymbol{x}^* + \boldsymbol{n}, \tag{1}$$

where $\boldsymbol{y} \in \mathbb{R}^n$ is the observed measurement, $\boldsymbol{x}^* \in \mathbb{R}^d$ denotes the underlying true signal, $\boldsymbol{A} \in \mathbb{R}^{n \times d}$ is a linear forward measurement operator, and $\boldsymbol{n} \in \mathbb{R}^n$ is the measurement noise independent of signal $\boldsymbol{x}^*$. The goal of solving a linear inverse problem is to retrieve the underlying signal $\boldsymbol{x}^*$ given the observed measurement. In real-world scenarios, the problem is often ill-posed, i.e., the underlying signal is high dimensional compared to the number of observations. Formally, when $n < d$, the solution to the inverse problem is not unique. From a Bayesian perspective, the prior is given by a data representation of the underlying true signal, and the inverse problem can be solved by calculating the posterior using the measurement likelihood (Idier, 2013).

Recent developments in continuous normalizing flows have shown tremendous success in sampling from underlying complex high-dimensional distributions (Chen et al., 2018). Particularly, flow matching models (Lipman et al., 2023; Liu et al., 2023b; Albergo & Vanden-Eijnden, 2023; Albergo et al., 2023; Ma et al., 2024) together with diffusion models (Song et al., 2021b; Ho et al., 2020; Song et al., 2021a; Rombach et al., 2022), emerge as powerful tools for image synthesis (Dhariwal & Nichol, 2021). The idea behind both models is to simulate a process that gradually perturbs data until it becomes random noise. Moving from data to noise, usually referred to as the forward process, can be represented by either a stochastic differential equation (SDE) or an ordinary differential equation (ODE) (Song et al., 2021b; Lipman et al., 2023). A score or velocity network is trained using the forward process to "memorize" the dynamics. Then, one can generate new data from noise by time-reversing the learned dynamics, which is usually referred to as the reverse process. In practice, velocity and score networks can be accurately learned so that the generation quality of flow matching and diffusion models is high, especially for image data.

---

[*]Equal Contribution in Alphabetical Order.

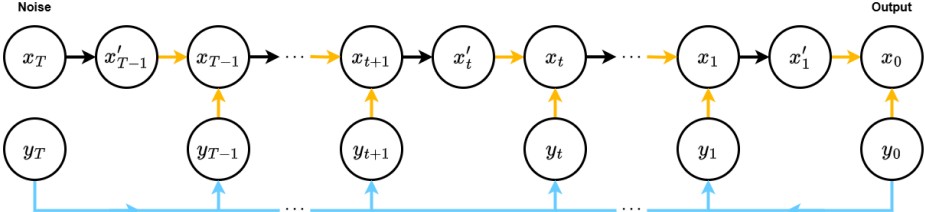

Figure 1: Overview of our FIG algorithm during the conditional sampling process. Black arrows ($\rightarrow$) denote the unconditional update. Orange arrows ($\rightarrow$) represent $K$ times conditional updates with unconditional sample $\boldsymbol{x}'_t$ and measurement interpolant $\boldsymbol{y}_t$ at each timestep $t$. Blue arrows ($\rightarrow$) indicate the measurement interpolation.

Due to the aforementioned advantages, flow matching and diffusion models have frequently been adopted to solve inverse problems for imaging data (Jalal et al., 2021). Specifically, for linear inverse problems, recent works use pre-trained flow matching or diffusion models as a prior and guide it towards the posterior by incorporating the measurement likelihood. Though existing methods achieve highly competitive results, most of them are either slow in inference or struggle in challenging tasks.

In this work, we propose a simple yet effective algorithm for linear inverse problems named "Flow with Interpolant Guidance" (FIG). We summarize its mechanism in Fig. 1. In FIG, the measurement $\boldsymbol{y}$ is interpolated parallel to the forward process to obtain measurement interpolants $\{\boldsymbol{y}_t\}_{t\in[0,T]}$. Then, the likelihood of $\boldsymbol{y}_t$ is used to perturb the reverse process. Our key contributions are summarized as follows:

- We propose a novel algorithm FIG that applies to all linear forward measurement operators and pre-trained models irrespective of complexity.
- We show that the updating scheme of FIG is theoretically justified.
- We implement FIG using both flow matching and diffusion models to solve various linear image reconstruction tasks, including challenging scenarios such as high measurement noise and severe ill-posedness. Results demonstrate that FIG achieves state-of-the-art performance with competitive runtime and memory consumption.

## 2 BACKGROUND AND RELATED WORKS

### 2.1 FLOW MATCHING AND DIFFUSION MODELS

Though various formulations were proposed, flow matching generally defines a continuous-time stochastic process $\{\boldsymbol{x}_t\}_{t\in[0,T]}$ for some $T \in (0,\infty]$, where $\boldsymbol{x}_0 = \boldsymbol{x}^*$ refers to the data distribution that one aims to draw samples from, and $\boldsymbol{x}_T = \boldsymbol{\varepsilon}_x$ is a noise variable that is easy to obtain. The marginal distribution of $\boldsymbol{x}_t$ for $t \in (0,T)$ is defined as an interpolant between the two boundaries, i.e.,

$$\boldsymbol{x}_t = \alpha_t \boldsymbol{x}_0 + \sigma_t \boldsymbol{x}_T, \tag{2}$$

where $\{\alpha_t\}_{t\in[0,T]}$ and $\{\sigma_t\}_{t\in[0,T]}$ are two smooth interpolation coefficients such that $\alpha_0 = \sigma_T = 1$ and $\alpha_T = \sigma_0 = 0$ (Albergo & Vanden-Eijnden, 2023; Ma et al., 2024). For readability, we remove the dependence of $\boldsymbol{x}_t$ on $t$ when there is no ambiguity. The interpolation Eq. (2) is the forward process of flow matching, whose dynamics can be represented by the probability flow ODE (Ma et al., 2024)

$$\dot{\boldsymbol{x}} = \boldsymbol{v}_t(\boldsymbol{x}). \tag{3}$$

Here, the velocity field $\boldsymbol{v}$ satisfies

$$\boldsymbol{v}_t(x) = \mathbb{E}(\dot{\boldsymbol{x}}_t | \boldsymbol{x}_t = x) = \dot{\alpha}_t \mathbb{E}(\boldsymbol{x}_0 | \boldsymbol{x}_t = x) + \dot{\sigma}_t \mathbb{E}(\boldsymbol{x}_T | \boldsymbol{x}_t = x). \tag{4}$$

Typically, the velocity is parameterized by a neural net $\boldsymbol{v}_\theta$ and trained by minimizing the loss

$$\mathcal{L}_{\boldsymbol{v}}(\theta) = \int_0^T \mathbb{E} \| \boldsymbol{v}_{t,\theta}(\boldsymbol{x}) - \dot{\alpha}_t \boldsymbol{x}_0 + \dot{\sigma}_t \boldsymbol{x}_T \|^2 dt. \tag{5}$$

Given a sufficiently well trained $\boldsymbol{v}_\theta$, one can sample the target $\boldsymbol{x}^*$ by numerically solving Eq. (3) in reverse time, i.e.,

$$\dot{\boldsymbol{x}} = -\boldsymbol{v}_t(\boldsymbol{x}), \tag{6}$$

with initial condition $\boldsymbol{x}_T = \boldsymbol{\varepsilon}_x$. This corresponds to the reverse process of flow matching.

In contrast to flow matching, diffusion models directly define the dynamics of the forward process through an SDE. Given certain choices of the drift and diffusion coefficients, the forward process converges to a noise distribution. More specifically, the forward diffusion model corresponds to the Ito process

$$d\boldsymbol{x} = \boldsymbol{f}(\boldsymbol{x}, t)dt + g(t)d\boldsymbol{W}, \tag{7}$$

where $\boldsymbol{f}$ and $g$ are predetermined, and $\boldsymbol{W}$ denotes standard Brownian motion. Specific choices of $\boldsymbol{f}$ and $g$ lead to different diffusion models. For example, taking $\boldsymbol{f}(\boldsymbol{x}, t) = -\frac{1}{2}\beta(t)\boldsymbol{x}$ and $g(t) = \sqrt{\beta(t)}$ for some $\beta(t) > 0$, we obtain the variance-preserving (VP) SDE (Song et al., 2021b; Ho et al., 2020)

$$d\boldsymbol{x} = -\frac{1}{2}\beta(t)\boldsymbol{x}dt + \sqrt{\beta(t)}d\boldsymbol{W}. \tag{8}$$

When $T \to \infty$, the noise $\boldsymbol{x}_T \sim \mathcal{N}(0, \boldsymbol{I})$.

The reverse process of Eq. (7) was shown (Anderson, 1982) to correspond to

$$d\boldsymbol{x} = \left[\boldsymbol{f}(\boldsymbol{x}, t) - g(t)^2 \boldsymbol{s}_t(\boldsymbol{x})\right] dt + g(t)d\overline{\boldsymbol{W}}, \tag{9}$$

where $\overline{\boldsymbol{W}}$ is a Brownian motion independent of $\boldsymbol{W}$, and $\boldsymbol{s}_t(\boldsymbol{x})$ denotes the score function of $\boldsymbol{x}_t$. Using the forward process defined in Eq. (8), one trains a neural net to learn the score function $\boldsymbol{s}_{t,\theta}(\boldsymbol{x}) \approx \nabla_x \log p(\boldsymbol{x}, t)$ via a loss similar to Eq. (5), where $p(\boldsymbol{x}, t)$ is the PDF of $\boldsymbol{x}_t$. One then runs the reverse process detailed in Eq. (9) with the trained score to obtain a data sample (Song et al., 2021b).

## 2.2 RELATION BETWEEN FLOW MATCHING AND DIFFUSION MODELS

There are both common grounds and differences between flow matching and diffusion models. For the flow matching setting, Ma et al. (2024) shows that

$$\boldsymbol{v}_t(\boldsymbol{x}) = a_t \boldsymbol{x} + b_t \boldsymbol{s}_t(\boldsymbol{x}), \tag{10}$$

where $a_t$, $b_t$ are determined by $\alpha_t$ and $\sigma_t$ in Eq. (2), demonstrating the fact that score and velocity are interchangeable. However, it can also be shown by Tweedie's formula (Efron, 2011) that

$$\boldsymbol{s}_t(\boldsymbol{x}_t) = -\alpha_t^{-1}\mathbb{E}(\boldsymbol{x}_0|\boldsymbol{x}_t), \tag{11}$$

which reveals a singularity in the score function at time $T$, since $\alpha_T = 0$. As $\boldsymbol{v}$ does not explode so long as the interpolation coefficients are smooth, training a velocity via flow matching is numerically more stable than training the score.

Though score and velocity are closely related, diffusion and flow matching models still differ significantly in terms of dynamics. In flow matching, there is only one noise variable $\boldsymbol{\varepsilon}_x$. Interpolation between data and noise leaves the path-wise trajectory of $\boldsymbol{x}_{[0,T]}$ smooth in $t$. In contrast, diffusion models are driven by Brownian motion, such that the path-wise trajectory is non-differentiable. Therefore, the law of a diffusion process and a flow matching process can never agree. Nevertheless, with certain choices of coefficients, a flow matching model and a diffusion model can share the same marginal distributions. For example, consider the VP SDE given in Eq. (8). Let $T = \infty$, $\alpha_t = \sqrt{1 - e^{-\int_0^t \beta(s)ds}}$, and $\sigma_t = e^{-\frac{1}{2}\int_0^t \beta(s)ds}$. Then the distribution of $\boldsymbol{x}_t$ generated by Eq. (2) is exactly the same as the one of the VP SDE given in Eq. (8). This suggests that a number of algorithms such as DPS (Chung et al., 2023b) can replace their diffusion based priors with flow matching priors, since it is often the marginal distribution instead of the law of the whole process that matters.

## 2.3 LINEAR INVERSE PROBLEMS WITH FLOW MATCHING AND DIFFUSION PRIOR

Recent attempts addressing linear inverse problems using flow matching or diffusion models can be classified into two categories: 1) task-specific methods that train a conditional diffusion model (Saharia et al., 2023; 2022; Whang et al., 2022); and 2) task-agnostic methods that only rely on

pre-trained unconditional diffusion models. For task specific methods, Liu et al. (2023a); Chung et al. (2023c) formulate the conditional diffusion as a Schrödinger bridge problem, which can be further related to the Doob's $h$-transform (Särkkä & Solin, 2019; Zhang et al., 2021) and stochastic optimal control (Uehara et al., 2024).

Task-agnostic methods tackle Bayesian inverse problems, where pre-trained flow matching or diffusion models are treated as priors. In contrast to the usual Bayesian settings, the prior can only be sampled by solving the reverse process. The key question is how to incorporated measurement likelihood into the reverse process such that it leads to a solution of the problem. Ideally, one method is to "hijack" the reverse process by adding a guidance such that the process is conditioned on the measurement $\boldsymbol{y}$. For instance, for diffusion priors, one aims to sample from the following SDE:

$$\mathrm{d}\boldsymbol{x} = \left[\boldsymbol{f}(\boldsymbol{x}, t) + g(t)^2 \nabla_{\boldsymbol{x}} \log p_t\left(\boldsymbol{x}|\boldsymbol{y}\right)\right] \mathrm{d}t + g(t)\mathrm{d}\boldsymbol{W}. \tag{12}$$

Since $\nabla_{\boldsymbol{x}_t} \log p_t\left(\boldsymbol{x}_t|\boldsymbol{y}\right) = \nabla_{\boldsymbol{x}} \log p_t\left(\boldsymbol{x}_t\right) + \nabla_{\boldsymbol{x}} \log q\left(\boldsymbol{y}|\boldsymbol{x}_t\right)$ (Bayes rule), $\nabla_{\boldsymbol{x}} \log q\left(\boldsymbol{y}|\boldsymbol{x}_t\right)$ is the guidance leading to the posterior. However, the conditional density $q\left(\boldsymbol{y}|\boldsymbol{x}_t\right)$ is intractable, because the measurement $\boldsymbol{y}$ only depends on $\boldsymbol{x}_0 = \boldsymbol{x}^*$ due to the linear inverse problem specified in Eq. (1).

Though the likelihood is intractable, various algorithms have been proposed to tackle linear inverse problems with diffusion or flow matching prior. Among the existing methods, Chung & Ye (2022); Chung et al. (2022); Zhu et al. (2023) adopt a projection onto the measurement subspace; Kawar et al. (2022); Meng & Kabashima (2022) approximate $q(\boldsymbol{y}|\boldsymbol{x}_t)$ via SVD; Mardani et al. (2024); Feng et al. (2023) use variational inference; Fei et al. (2023); Chung et al. (2023b); Song et al. (2023); Pokle et al. (2024); Pandey et al. (2024) estimate the likelihood score $\nabla_{\boldsymbol{x}_t} \log q(\boldsymbol{y}|\boldsymbol{x}_t)$; Dou & Song (2024); Trippe et al. (2023); Cardoso et al. (2024) carry out exact Bayesian posterior estimation with sequential Monte Carlo (SMC); Choi et al. (2021); Song et al. (2022) enforce data consistency via optimization at each time step; Ben-Hamu et al. (2024) optimize the starting noise so that the generation process yields the intended results; Xu & Chi (2024); Li et al. (2024) use the variable splitting techniques (Geman & Yang, 1995; Afonso et al., 2010; Boyd et al., 2011; Venkatakrishnan et al., 2013) to decouple the optimization with respect to prior and data consistency; Zhang et al. (2024) introduce a decoupled noise annealing process to handle highly nonlinear problems. Despite promising experimental results, we find prior works struggle to efficiently handle challenging linear inverse problems. More specifically, we find methods of high reconstruction quality are often time consuming, and efficient methods fail to consistently obtain high quality reconstruction throughout different tasks. Furthermore, current methods struggle to effectively handle high levels of measurement noise.

## 3 METHOD

In this section, we propose 'Flow with Interpolant Guidance' (FIG), a simple yet effective algorithm that can leverage all pre-trained flow matching and diffusion models to solve linear inverse problems. During initialization, we perturb the measurement $\{\boldsymbol{y}\}_{t\in[0,T]}$ parallel to the forward process of $\boldsymbol{x}$ to get measurement interpolants $\boldsymbol{y}_t$ similar to (Song et al., 2022). The intuition of measurement interpolants comes from filtering (Dou & Song, 2024). With the interpolants, the likelihood $q_t(\boldsymbol{y}_t|\boldsymbol{x}_t)$ is Gaussian for all $t$. Then, at each time $t \in [0, T]$, we incorporate $q_t(\boldsymbol{y}_t|\boldsymbol{x}_t)$ as guidance for sampling $\boldsymbol{x}_t$. This section is organized as follows. In Section 3.1, we state how to obtain measurement interpolants. Section 3.2 presents the FIG algorithm. We then show in Section 3.3 that the update scheme is theoretically justified. Section 3.4 discusses details regarding the practical implementation of our algorithm. For readability, we choose to adopt a flow matching narrative in most of this section, although FIG can also use a diffusion prior. Empirical results using both flow matching and diffusion priors are shown in Section 4.

### 3.1 GENERATING MEASUREMENT INTERPOLANTS

Given a pre-trained flow model, $\boldsymbol{x}_t$ is interpolated following Eq. (2), with $\boldsymbol{x}_0$ following the data distribution and $\boldsymbol{x}_T = \boldsymbol{\varepsilon}_x \sim \mathcal{N}(0, \boldsymbol{I})$. We consider a parallel interpolation of $\boldsymbol{y}$, i.e.,

$$\boldsymbol{y}_t = \alpha_t \boldsymbol{y}_0 + \sigma_t \boldsymbol{y}_T, \tag{13}$$

where $\boldsymbol{y}_0 = \boldsymbol{y}$, and $\boldsymbol{y}_T = \boldsymbol{\varepsilon}_y = \boldsymbol{A}\boldsymbol{\varepsilon}_x$. The following sequence of equalities reveals the distribution of $q_t(\boldsymbol{y}_t|\boldsymbol{x}_t)$:

$$\boldsymbol{y}_t = \alpha_t(\boldsymbol{A}\boldsymbol{x}_0 + \boldsymbol{n}) + \sigma_t \boldsymbol{A}\boldsymbol{\varepsilon}_x = \boldsymbol{A}(\alpha_t \boldsymbol{x}_0 + \sigma_t \boldsymbol{x}_T) + \alpha_t \boldsymbol{n} = \boldsymbol{A}\boldsymbol{x}_t + \alpha_t \boldsymbol{n}. \tag{14}$$

---

**Algorithm 1** Flow with Interpolant Guidance (FIG)

---

**Require:** $T, c, K, w, \boldsymbol{y}_0$
 1: Initialize $\boldsymbol{x}_T = \boldsymbol{\varepsilon}_x \sim \mathcal{N}(\boldsymbol{0}, \boldsymbol{I})$ ▷ Initialize $\boldsymbol{x}_t$
 2: $\Delta t = 1/T$
 3: **for** $i = T$ **to** $1$ **do**
 4:     $t = i/T, t' = (i-1)/T$
 5:     $\boldsymbol{y}_{i-1} = \alpha_{t'} \boldsymbol{y}_0 + w\sigma_{t'} \boldsymbol{A}\boldsymbol{\varepsilon}_x$ ▷ measurement interpolation with rescaled variance
 6:     $\boldsymbol{x}_{i-1} = \boldsymbol{x}_i - \boldsymbol{v}_\theta(\boldsymbol{x}_i, t)\Delta t$ ▷ Unconditional update
 7:     **for** $k = 1$ **to** $K$ **do**
 8:         $\boldsymbol{x}_{i-1} = \boldsymbol{x}_{i-1} - \frac{c\lambda_t \sigma_t \Delta t}{2\alpha_t^2 \sigma_n^2} \nabla_{\boldsymbol{x}_{i-1}} \|\boldsymbol{y}_{i-1} - \boldsymbol{A}\boldsymbol{x}_{i-1}\|_2^2$ ▷ $K$-steps conditional update
 9:     **end for**
10: **end for**

---

Hence, we have $q_t(\boldsymbol{y}_t | \boldsymbol{x}_t) = \mathcal{N}\left(\boldsymbol{y}_t; \boldsymbol{A}\boldsymbol{x}_t, \alpha_t^2 \sigma_n^2 \boldsymbol{I}\right)$. We note that the generation of the process $\boldsymbol{y}_t$ relies on the fact that the measurement $\boldsymbol{y}_0$ has a linear relationship with the signal $\boldsymbol{x}_0$, therefore the interpolant measurement only works for linear models.

It is worth mentioning that $\boldsymbol{x}_t$ in Eq. (14) is defined by the interpolation process given in Eq. (2), instead of the probability flow ODE detailed in Eq. (3). In this case, it is fair to ask: is the conditional distribution of $\boldsymbol{y}_t | \boldsymbol{x}_t$ still Gaussian when $\boldsymbol{x}_t$ is sampled from Eq. (3)? The answer is yes because $\boldsymbol{x}_t$ defined in Eq. (2) and Eq. (3) have the same distribution (Liu et al., 2023b).

## 3.2 ALGORITHM OVERVIEW

Before diving into the detailed analysis, we first provide an overview of the FIG algorithm (see Algorithm 1). The algorithm numerically solves a reverse-time ODE to generate samples from the correct posterior,

$$d\boldsymbol{x} = -\boldsymbol{v}_t(\boldsymbol{x})dt + \lambda_t \sigma_t \nabla_x \log q_t(\boldsymbol{y}_t | \boldsymbol{x})dt, \quad \text{with} \quad \boldsymbol{x}_T \sim \mathcal{N}(0, \boldsymbol{I}). \tag{15}$$

Here, $\boldsymbol{v}_t$ denotes the (unconditional) velocity field from the pre-trained model, $\lambda_t = \dot{\sigma}_t - \frac{\dot{\alpha}_t}{\alpha_t}\sigma_t$, and $\nabla_x \log q_t(\boldsymbol{y}|\boldsymbol{x}) = -\frac{1}{2\alpha_t^2 \sigma_n^2} \nabla_x \|\boldsymbol{y} - \boldsymbol{A}\boldsymbol{x}\|^2$. The derivation and justification of Eq. (15) are detailed in Section 3.3.

Numerically, time is discretized on a uniform grid with $\Delta t$ referring to the step size. In the following, we use $\boldsymbol{x}_i$ and $\boldsymbol{y}_i$ instead of $\boldsymbol{x}_{t_i}$ and $\boldsymbol{y}_{t_i}$ for readability. The updating scheme at each time $t_i$, corresponding to an Euler's method with splitting (Leimkuhler & Matthews, 2015), is as follows:

$$\begin{cases} \boldsymbol{x}'_{i-1} & = \boldsymbol{x}_i - \boldsymbol{v}_{t_i}(\boldsymbol{x}_i)\Delta t \\ \boldsymbol{x}_{i-1} & = \boldsymbol{x}'_{i-1} - \frac{\lambda_t \sigma_t}{2\alpha_t^2 \sigma_n^2} \nabla_{x'_{i-1}} \|\boldsymbol{y}_{i-1} - \boldsymbol{A}\boldsymbol{x}'_{i-1}\|^2 \Delta t. \end{cases} \tag{16}$$

Each step of the two-step updating scheme has a different aim. The first step in Eq. (16) corresponds to the unconditional update, which is exactly a numerical discretization of the reverse process specified in Eq. (6). In practice, it is performed via Euler's method (see line 6 in Algorithm 1). The second step is a conditional update that incorporates measurement information. In practice $K$ gradient descent steps with learning rate $c$ carry out the conditional update (see line 7 to line 9 in Algorithm 1). We next introduce the theoretical reasoning for this simple but effective updating scheme.

## 3.3 THEORETICAL JUSTIFICATION

We now provide the theoretical reasoning underlying FIG. Importantly, note that for all $t \in (0, T)$, the marginal distribution of $\boldsymbol{y}_t$ is uniquely determined by $\boldsymbol{y}_0$ and $\boldsymbol{\varepsilon}_y$ as detailed in Eq. (13). We make the following technical assumption:

**Assumption 1** *The conditional distribution of $\boldsymbol{x}_t$ given $\boldsymbol{y}_0$ and $\boldsymbol{\varepsilon}_y$ is equivalent to the conditional distribution of $\boldsymbol{x}_t$ given $\boldsymbol{y}_t$.*

Let $p_t(\cdot|\boldsymbol{y}_t)$ be the density of $\boldsymbol{x}_t$ given $\boldsymbol{y}_t$. The following theorem reviews the dynamics of $p_t(\cdot|\boldsymbol{y}_t)$.

**Theorem 1** *Let $\boldsymbol{v}_t(\boldsymbol{x}|\boldsymbol{y}_t)$ be the conditional velocity field that generates $p_t$, i.e., $p_t(\boldsymbol{x}|\boldsymbol{y}_t)$ solves the reverse-time continuity equation with initial condition $p_T$ being a standard Gaussian:*

$$\partial_t p_t(\boldsymbol{x}|\boldsymbol{y}_t) - \nabla_x \cdot [\boldsymbol{v}_t(\boldsymbol{x}|\boldsymbol{y}_t)p_t(\boldsymbol{x}|\boldsymbol{y}_t)] = 0. \tag{17}$$

*Let $\boldsymbol{s}_t(\boldsymbol{x}|\boldsymbol{y}_t) = \nabla_x \log p_t(\boldsymbol{x}|\boldsymbol{y}_t)$ be the score function. Then under Assumption 1,*

$$\boldsymbol{v}_t(\boldsymbol{x}|\boldsymbol{y}_t) = \frac{\dot{\alpha}_t}{\alpha_t}\boldsymbol{x} - \lambda_t \sigma_t \boldsymbol{s}_t(\boldsymbol{x}|\boldsymbol{y}_t). \tag{18}$$

We defer the proof to Appendix A.1. Theorem 1 describes the conditional dynamics, which demonstrates consistency, i.e., posterior sampling will be achieved if the conditional continuity equation is solved in reverse time. The corresponding (reverse-time) probability flow ODE gives rise to the numerical updating scheme Eq. (16):

$$d\boldsymbol{x} = -\boldsymbol{v}_t(\boldsymbol{x}|\boldsymbol{y}_t)dt, \tag{19}$$

with initial condition $\boldsymbol{x}_T \sim \mathcal{N}(0, \boldsymbol{I})$. In view of Eq. (18), we rewrite the conditional probability flow ODE detailed in Eq. (19) as

$$\begin{aligned}
d\boldsymbol{x} = -\boldsymbol{v}_t(\boldsymbol{x}|\boldsymbol{y}_t)dt &= -\Big(\frac{\dot{\alpha}_t}{\alpha_t}\boldsymbol{x} - \lambda_t \sigma_t \nabla_x \log p_t(\boldsymbol{x}|\boldsymbol{y}_t)\Big)dt \\
&= -\Big(\frac{\dot{\alpha}_t}{\alpha_t}\boldsymbol{x} - \lambda_t \sigma_t \nabla_x \log p_t(\boldsymbol{x})\Big)dt + \lambda_t \sigma_t \nabla_x \log q_t(\boldsymbol{y}_t|\boldsymbol{x})dt \\
&= -\boldsymbol{v}_t(\boldsymbol{x})dt + \lambda_t \sigma_t \nabla_x \log q_t(\boldsymbol{y}_t|\boldsymbol{x})dt.
\end{aligned} \tag{20}$$

Here, $\nabla_x \log q_t(\boldsymbol{y}|\boldsymbol{x}) = -\frac{1}{2\alpha_t^2 \sigma_n^2}\nabla_x \|\boldsymbol{y} - \boldsymbol{A}\boldsymbol{x}\|^2$. Further, $\boldsymbol{v}_t$ denotes the (unconditional) velocity field from the pre-trained model. Note that this equation is identical to Eq. (15). An Euler discretization of Eq. (15) yields exactly Eq. (16). Interestingly, when we let $\alpha_t = 1 - t$ and $\sigma_t = t$, i.e., when we follow the rectified flow (Liu et al., 2023b), the coefficient $\lambda_t \sigma_t = \frac{t}{1-t} = 1/\text{SNR}_t$, where $\text{SNR}_t = \frac{1-t}{t} = \frac{\alpha_t}{\sigma_t}$. The following corollary summarizes the direction of the conditional update.

**Corollary 1** *The conditional update direction at time $t_i$ is equivalent to a gradient flow direction that maximizes an $\mathcal{L}_2$-regularized posterior log-likelihood at the next time-step $\log p_{i-1}(\boldsymbol{x}_{i-1}|\boldsymbol{y}_{i-1})$:*

$$\nabla_{x_{i-1}}\Big(\log p_{i-1}(\boldsymbol{x}_{i-1}|\boldsymbol{y}_{i-1}) + \frac{|\dot{\alpha}_{i-1}|}{\alpha_{i-1}}\frac{1}{SNR_t}\frac{\|\boldsymbol{x}_{i-1}\|^2}{2}\Big). \tag{21}$$

Corollary 1 demonstrates that FIG updating direction is equivalent to maximizing a regularized log-likelihood, providing further intuition into the algorithm. The proof is provided in Appendix A.2.

While all previous derivations use a flow matching base model, FIG can also be applied to diffusion models. When adopting a diffusion prior, the following corollary states the SDE version of FIG.

**Corollary 2** *Let the pre-trained unconditional generation be a diffusion model, as detailed in Eq. (7), then FIG admits the form:*

$$\mathrm{d}\boldsymbol{x} = \big[\boldsymbol{f}(\boldsymbol{x}, t) + g(t)^2 \boldsymbol{s}_t(\boldsymbol{x}|\boldsymbol{y}_t)\big]\,\mathrm{d}t + g(t)\mathrm{d}\overline{\boldsymbol{W}}. \tag{22}$$

Numerically solving Eq. (22) gives the FIG algorithm with diffusion prior. It is interesting to observe that Eq. (22) coincides with the limiting process of FPS (Dou & Song, 2024), yet the sampling procedure of FIG remains different. The proof is given in Appendix A.3.

## 3.4 PRACTICAL IMPLEMENTATION

In practice, we observe that a conditional update can be viewed as minimizing the distance between the data $\boldsymbol{x}_t$ projected onto the measurement space and the measurement interpolants $\boldsymbol{y}_t$. The likelihood score represents the gradient of the conditional update. Therefore, a single conditional update corresponds to one step of gradient descent. If we perform only one conditional update per time-step and adjust the magnitude of the update solely by tuning the weight $c$ of the likelihood score, the effectiveness of the conditional update within a single time-step will be significantly

| Method | SR ($4\times$) | | | Gaussian Deblur | | | Motion Deblur | | | Inpainting | | |
|---|---|---|---|---|---|---|---|---|---|---|---|---|
| | PSNR ↑ | SSIM ↑ | LPIPS ↓ | PSNR ↑ | SSIM ↑ | LPIPS ↓ | PSNR ↑ | SSIM ↑ | LPIPS ↓ | PSNR ↑ | SSIM ↑ | LPIPS ↓ |
| FIG-Flow | **29.41** | **0.811** | **0.201** | **26.83** | **0.738** | 0.261 | **30.84** | **0.871** | **0.209** | **27.08** | **0.814** | 0.274 |
| DPS-Flow | 28.73 | 0.777 | 0.221 | 22.95 | 0.568 | 0.337 | 21.95 | 0.571 | 0.382 | 27.05 | 0.801 | **0.244** |
| DMPS | 29.00 | 0.807 | 0.222 | 26.74 | 0.725 | 0.255 | 29.81 | 0.828 | 0.226 | 12.66 | 0.284 | 0.658 |
| OT-ODE | 28.40 | 0.769 | 0.225 | 26.34 | 0.708 | **0.252** | 29.07 | 0.818 | 0.226 | 18.86 | 0.475 | 0.500 |
| FIG-Diffusion | **29.89** | **0.846** | 0.163 | **29.83** | **0.845** | 0.147 | **31.81** | **0.882** | 0.137 | 27.36 | **0.848** | 0.163 |
| DPS-Diffusion | 28.85 | 0.801 | 0.172 | 28.32 | 0.797 | **0.136** | 30.69 | 0.843 | 0.127 | 28.23 | 0.793 | 0.164 |
| DDNM+ | 28.82 | 0.750 | 0.340 | 7.84 | 0.021 | 0.840 | - | - | - | **28.70** | 0.843 | 0.182 |
| DAPS | 29.59 | 0.809 | 0.158 | 29.74 | 0.808 | 0.146 | 31.67 | 0.859 | **0.119** | 27.85 | 0.791 | 0.183 |

Table 1: Quantitative comparison (PSNR, SSIM, LPIPS) of different algorithms for different tasks on the CelebA-HQ $256 \times 256$ test dataset. All input images have a measurement Gaussian noise of $\sigma_n = 0.05$. **Bold** for the best.

reduced. However, if this distance is fully optimized, the log likelihood of measurement interpolant will be overwhelming compared to the prior information, resulting in severe overfitting. Thus, maintaining a balance between the measurement information and the prior information is crucial for our reconstruction problem. To remedy the issue, we introduce a new parameter $K$. After an unconditional update, we perform $K$ conditional updates with theoretically justified weights and directions as previously shown. We also note that due to the nature of ODEs, once the velocity field and initialization are determined, the final generated data is uniquely determined. However, the random initialization does not correspond to the desired reconstruction result. Therefore, we opt to average multiple random initializations, thereby reducing the impact of incorrect initial values during measurement interpolation. Given that our measurement model is linear, this approach is equivalent to introducing another parameter $w$ to generate $\boldsymbol{y}_t$, i.e., the new $\boldsymbol{y}_t$ is generated by $\alpha_t \boldsymbol{y}_0 + w \sigma_t \boldsymbol{\varepsilon}_y$.

By combining the practical design considerations, we obtain Algorithm 1. The parameters $c$, $w$, $K$ are constants, with $c$ and $K$ being task-specific, governing the balance between unconditional and conditional updates. See Appendix B for more details on the choice of hyper-parameters. The parameter $w$ helps us obtain an empirically better log-likelihood of the measurement interpolants. The effectiveness of both strategies is examined in later ablation studies.

Although FIG is effective in solving most linear imaging inverse problems, it struggles with inpainting tasks where a large region is masked. The reason is that the masked pixels carry zero information for the conditional update, as they are totally "masked". Therefore, we propose a simple remedy termed FIG+. It improves the performance for inpainting tasks. Concretely, for each $\boldsymbol{x}_t$ update, we first estimate $\hat{\boldsymbol{x}}_0$ using Tweedie's formula, and perturb it to time $t$ to get $\hat{\boldsymbol{x}}_t$. Then we mix the masked parts of $\hat{\boldsymbol{x}}_t$ with the current $\boldsymbol{x}_t$ to update the masked region. A hyper-parameter mixing weight $m \in [0, 1]$ determines how much we trust $\hat{\boldsymbol{x}}_0$. We provide the pseudocode in Appendix C. When $m = 0$, we get the original FIG algorithm. The additional three steps of FIG+ only involve simple addition and matrix (mask) multiplication. Hence they don't affect the runtime and memory consumption much. In our experiments below, we only use FIG+ for the inpainting task.

## 4 EXPERIMENTS

### 4.1 EXPERIMENTAL SETUP

**Datasets.** We conduct experiments on 3 natural image datasets: CelebA-HQ (Karras et al., 2018), LSUN-Bedroom (Yu et al., 2015), and AFHQ-Cat (Choi et al., 2020). All images are taken from the official test data splits and are preprocessed to the size of $256 \times 256 \times 3$.

**Tasks.** Our evaluation focuses on linear inverse problems including super-resolution, deblurring, and inpainting. For super-resolution, we apply $4\times$ bicubic downsampling across all datasets. For deblurring tasks, we use Gaussian blurring with a kernel size of $61 \times 61$ and a standard deviation of 3.0, and motion blurring with the same kernel size but a standard deviation of 0.5. For inpainting, we perform random inpainting by masking out 90% of the total pixels. We follow the SVD-based measurement operations defined in DDRM (Kawar et al., 2022), as DDNM, DMPS, and OT-ODE

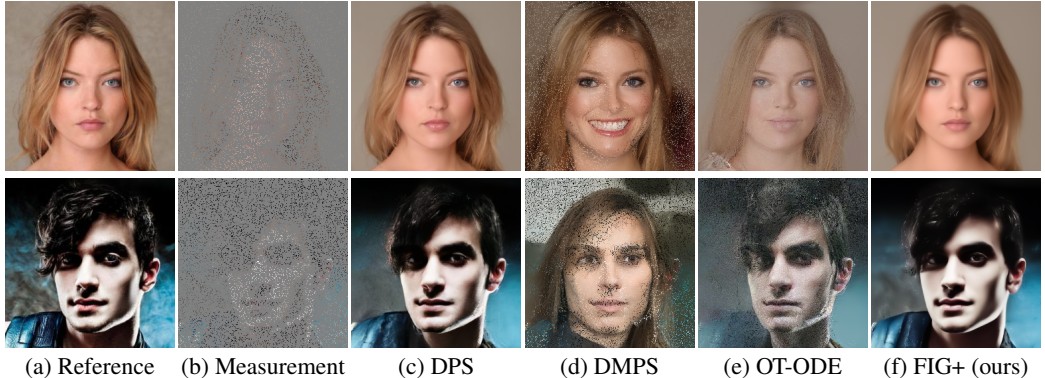

(a) Reference    (b) Measurement    (c) DPS    (d) DMPS    (e) OT-ODE    (f) FIG+ (ours)

Figure 2: Results for 90% random inpainting with noise $\sigma_n = 0.05$ on the CelebA-HQ dataset using a flow matching base model.

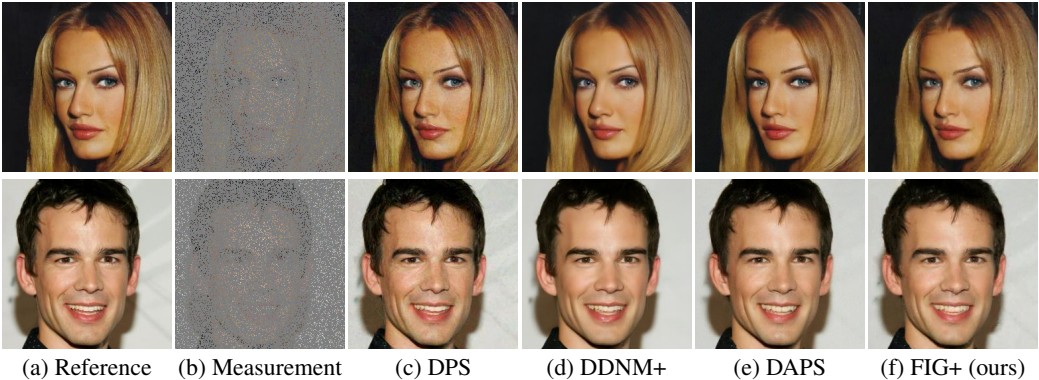

(a) Reference    (b) Measurement    (c) DPS    (d) DDNM+    (e) DAPS    (f) FIG+ (ours)

Figure 3: Results for 90% random inpainting with noise $\sigma_n = 0.05$ on the CelebA-HQ dataset using a diffusion base model.

require an SVD for their conditional updates. For all tasks above, we add a measurement Gaussian noise $\boldsymbol{n} \sim \mathcal{N}(\boldsymbol{0}, \sigma_n^2 \boldsymbol{I})$ with $\sigma_n = 0.05$. To further demonstrate the capabilities of our algorithm, we conducted two additional, more complex experiments: $4\times$ super-resolution with high noise ($\sigma_n = 1.0$), $16\times$ super-resolution with noise ($\sigma_n = 0.2$).

**Metrics.** For the quantitative comparison, we use the perceptual Learned Perceptual Image Patch Similarity (LPIPS) distance (Zhang et al., 2018), along with two standard metrics: peak signal-to-noise-ratio (PSNR), and structural similarity index (SSIM).

**Baselines.** We compare the performance of our algorithm to several state-of-the-art training-free algorithms for solving inverse problems, including DPS (Chung et al., 2023b), OT-ODE (Pokle et al., 2024), DMPS (Meng & Kabashima, 2022), DDNM/DDNM+ (Wang et al., 2023), and DAPS (Zhang et al., 2024). OT-ODE is an improved version of ΠGDM (Song et al., 2023) applied to flow matching models, thus it is used instead of ΠGDM as the baseline. Likewise, we transferred DPS and DMPS to flow matching using Eq. (18). Leveraging the advantages of flow matching models, we fine-tune the baseline methods to ensure they all achieve their best performance at 50 NFEs except for OT-ODE. It requires less than 50 NFEs due to its start time strategy.

**Base Models.** We implemented our FIG algorithm for both flow matching and diffusion models. We refer to both incarnations via FIG-Flow and FIG-Diffusion. For each base model, we then compared with three state-of-the-art algorithms respectively. For FIG-Flow, we use the pre-trained Rectified Flow model from Liu et al. (2023b) as our base model and implement all baselines on it to ensure a fair comparison. In the diffusion category, we utilize EDM (Karras et al., 2022) as the base model for our algorithm, DPS, and DAPS. Since DDNM/DDNM+ employs a unique time travel structure that may not transfer well, we retain their original DDIM base model and conduct experiments using their original code. For unconditional generation, we choose the Euler ODE solver for simplicity as it makes it easier to add the measurement information at each sampling step.

| Method | SR ($4\times$) $\sigma_n = 1.0$ | | | SR ($16\times$) $\sigma_n = 0.2$ | | |
|---|---|---|---|---|---|---|
| | PSNR ↑ | SSIM ↑ | LPIPS ↓ | PSNR ↑ | SSIM ↑ | LPIPS ↓ |
| FIG (ours) | **24.47** | **0.722** | **0.315** | **19.92** | **0.553** | **0.405** |
| DPS | 22.50 | 0.593 | 0.378 | 19.20 | 0.531 | 0.424 |
| DMPS | 24.06 | 0.678 | 0.342 | 19.91 | 0.536 | 0.441 |
| OT-ODE | 15.14 | 0.191 | 0.695 | 17.79 | 0.408 | 0.570 |

Table 2: Quantitative comparison (PSNR, SSIM, LPIPS) of different algorithms for high noise bicubic super-resolution on the CelebA-HQ $256 \times 256$ test dataset. **Bold** for the best.

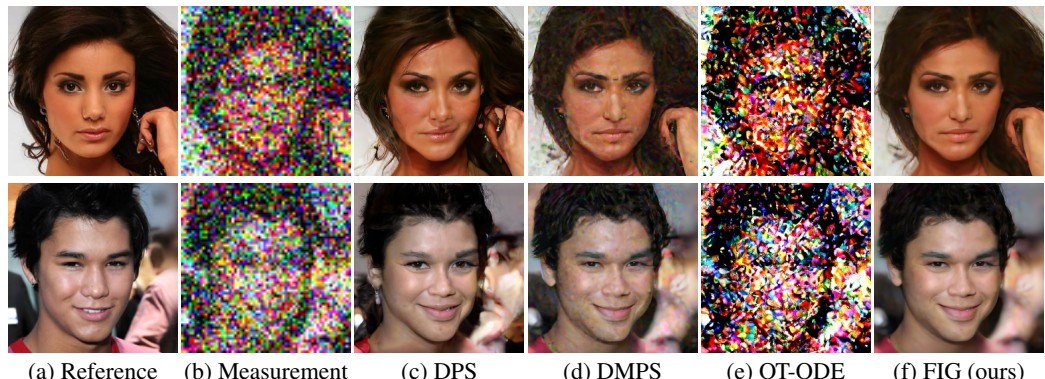

(a) Reference    (b) Measurement    (c) DPS    (d) DMPS    (e) OT-ODE    (f) FIG (ours)

Figure 4: Results for $4\times$ bicubic super-resolution with high noise $\sigma_n = 1.0$ on the CelebA-HQ dataset.

## 4.2 EXPERIMENTAL RESULTS

The quantitative comparisons are listed in Tables 1, 2, 11 and 12, with samples of reconstructed images shown in Figs. 2 to 6. Additional experimental results are presented in Appendices F and G. Our algorithm FIG demonstrates highly competitive performance across all tasks, metrics, and datasets. For all these tasks, DPS and OT-ODE tend to produce sharper edges, but this sometimes results in unrealistic details and textures, leading to worse metrics. In contrast, our algorithm generates faithfully smooth images with a good balance of detail and realism.

Our algorithm exhibits superior performance compared to other baselines with flow matching prior, particularly for the challenging super-resolution tasks with high levels of noise. We do not include the results from diffusion-based algorithms for those tasks because they all fail to yield a reasonable reconstruction. Note that Wang et al. (2023) reported a similar task with average pooling super-resolution but it fails on bicubic super-resolution. See Appendix E for more details of DDNM/DDNM+. Additionally, to further demonstrate the robustness of our algorithm, we test it on $4\times$ super-resolution with non-uniform and high noise. As shown in Fig. 6, our algorithm performs well even under these challenging conditions.

| Algorithm | Total Memory | NFEs | Avg. Inf. Time (s) |
|---|---|---|---|
| FIG-Flow | **7735MB** | 50 | **2.80±0.12** |
| DPS-Flow | 8109MB | 50 | 4.41±0.14 |
| DMPS | 7847MB | 50 | 2.83±0.15 |
| OT-ODE | 8257MB | 40 | 3.69±0.14 |
| FIG-Diffusion | **2809MB** | 100 | 4.06±1.01 |
| DPS-Diffusion | 7761MB | 100 | 8.19±1.04 |
| DDNM+ | 28707MB | 1210 | 28.36±0.61 |
| DAPS | **2809MB** | 1000 | 59.56±1.70 |

Table 3: The runtime and memory consumption required for different algorithms on the CelebA-HQ $256 \times 256$ test dataset for the super-resolution task. All experiments are conducted on a single NVIDIA RTX A6000 GPU for reconstructing one image. **Bold** for the best.

In addition to delivering outstanding performance, our algorithm maintains a low computational cost in terms of both runtime and memory usage as shown in Table 3.

To validate our theoretical results and the effectiveness of practical design components, we perform ablation studies (detailed in Appendix B) on 1) $K$-steps of conditional update with results in Fig. 7 and Tables 4 to 6 in Appendix B.1; and 2) measurement interpolant variance rescaling with results in

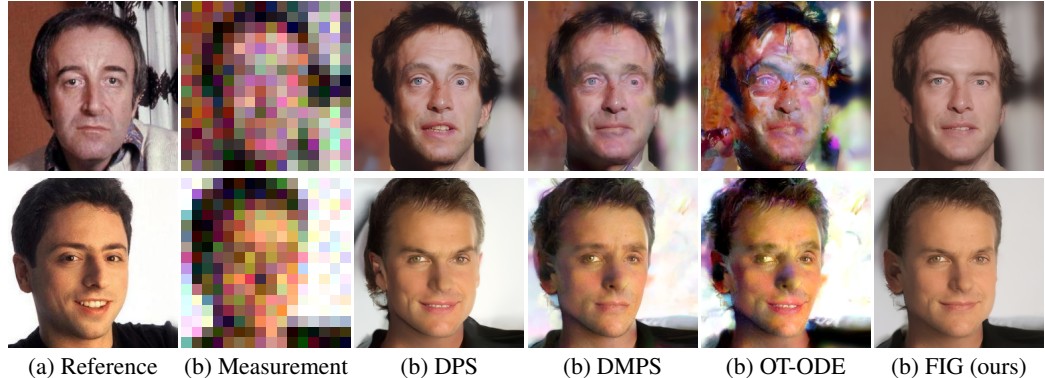

(a) Reference    (b) Measurement    (b) DPS    (b) DMPS    (b) OT-ODE    (b) FIG (ours)

Figure 5: Results for $16\times$ bicubic super-resolution with high noise $\sigma_n = 0.2$ on the CelebA-HQ dataset.

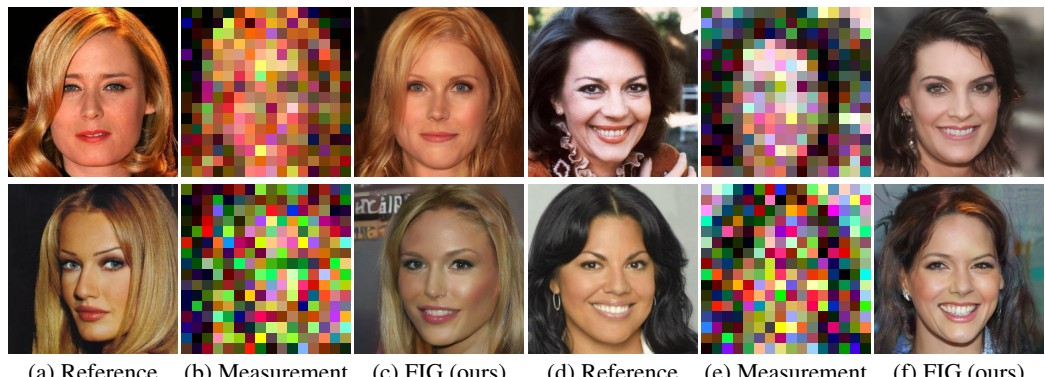

(a) Reference    (b) Measurement    (c) FIG (ours)    (d) Reference    (e) Measurement    (f) FIG (ours)

Figure 6: Results for $16\times$ bicubic super-resolution with high non-uniform Gaussian noise (first row) and non-uniform Laplacian noise (second row) on CelebA-HQ dataset.

Fig. 8 and Tables 7 to 9 in Appendix B.2. Both the visual outputs and quantitative results support the principles underlying the development of FIG.

## 5 CONCLUSION

In this paper, we presented FIG, a task-agnostic algorithm for solving linear inverse problems. Our key observation: adding noise to the measurement is easier compared to denoising the signal. We thus use measurement interpolants to guide the unconditional generation. The FIG algorithm is simple yet effective, as is shown by experiments on multiple tasks over several high-dimensional datasets.

Limitations: First, since the measurement interpolants rely on the linear relationship between the measurement and the underlying signal, FIG can not directly be applied to general non-linear inverse problems. For the same reason, the algorithm is not compatible with pre-trained models with latent encodings. Second, the FIG result is still an approximated posterior. To obtain true posterior samples, one should consider methods based on stochastic optimal control (Guo et al., 2024; Uehara et al., 2024). Another way is to sequentially update $\varepsilon_x$ so that it generates the intended image (Ben-Hamu et al., 2024).

Future Work: Although FIG can not be extended to non-linear problems in general, its potential application in bilinear problems such as blind motion deblurring (Chung et al., 2023a; Fei et al., 2023) could be an interesting direction. When fixing either the blurring kernel or the image, the noise model is linear, thus measurement interpolants can be constructed. Using FIG's updating scheme could potentially lead to better reconstruction. However, initialization and regularization on the kernel are highly non-trivial, and require future explorations. Another interesting direction is to extend FIG to latent models (Rout et al., 2024; Song et al., 2024).

**Acknowledgements:** YY, YZ and ZZ are supported in part by the NSF grant 1934757 and the Alfred P. Sloan Foundation. XM is supported by NSFC No. 62306277, and the Fundamental Research Funds for the Zhejiang Provincial Universities Grant No. K20240090.

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

APPENDIX: FIG: FLOW WITH INTERPOLANT GUIDANCE FOR LINEAR INVERSE PROBLEMS

## A PROOFS

### A.1 PROOF OF THEOREM 1

The proof follows from Appendix C of Ma et al. (2024). We start with the conditional density $\overline{p}_t(\boldsymbol{x}|\boldsymbol{y}_0, \boldsymbol{\varepsilon}_y)$ and later use Assumption 1 to get $p_t(\boldsymbol{x}|\boldsymbol{y}_t)$. Let

$$\hat{p}_t(k) = \int e^{ik \cdot \boldsymbol{x}} \overline{p}_t(\boldsymbol{x}|\boldsymbol{y}_0, \boldsymbol{\varepsilon}_y) dx \tag{23}$$

be the characteristic function, which is also equal to

$$\hat{p}_t(k) = \tilde{\mathbb{E}} e^{ik \cdot \boldsymbol{x}_t}. \tag{24}$$

Here $\tilde{\mathbb{E}}$ denote the expectation on $\boldsymbol{x}_0|\boldsymbol{y}_0, \boldsymbol{\varepsilon}_y$ and $\boldsymbol{\varepsilon}_x|\boldsymbol{y}_0, \boldsymbol{\varepsilon}_y$. Similar to Ma et al. (2024), we have

$$\begin{aligned}
\partial_t \hat{p}_t(k) &= \partial_t \tilde{\mathbb{E}} e^{ik \cdot \boldsymbol{x}_t} \\
&= ik \tilde{\mathbb{E}}(\dot{\boldsymbol{x}}_t e^{ik \cdot \boldsymbol{x}_t}) \\
&= ik \mathbb{E}_{\overline{p}_t} \tilde{\mathbb{E}}(\dot{\boldsymbol{x}}_t e^{ik \cdot \boldsymbol{x}_t}|\boldsymbol{x}_t) \\
&= ik \mathbb{E}_{\overline{p}_t} [\tilde{\mathbb{E}}(\dot{\boldsymbol{x}}_t|\boldsymbol{x}_t) e^{ik \cdot \boldsymbol{x}_t}] \\
&= ik \mathbb{E}_{\overline{p}_t} (\tilde{\boldsymbol{v}}_t(\boldsymbol{x}_t) e^{ik \cdot \boldsymbol{x}_t}),
\end{aligned} \tag{25}$$

where we define $\tilde{\boldsymbol{v}}_t(\boldsymbol{x}_t) = \tilde{\mathbb{E}}(\dot{\alpha}_t \boldsymbol{x}_0 + \dot{\sigma}_t \boldsymbol{\varepsilon}_x|\boldsymbol{x}_t) = \mathbb{E}(\dot{\alpha}_t \boldsymbol{x}_0 + \dot{\sigma}_t \boldsymbol{\varepsilon}_x|\boldsymbol{x}_t, \boldsymbol{y}_0, \boldsymbol{\varepsilon}_y)$.

Now using the two formulations of the characteristic function given in Eq. (23) and Eq. (24), we have

$$\partial_t \int e^{ik \cdot \boldsymbol{x}} \overline{p}_t(\boldsymbol{x}|\boldsymbol{y}_0, \boldsymbol{\varepsilon}_y) dx = ik \int \tilde{\boldsymbol{v}}_t(\boldsymbol{x}_t) e^{ik \cdot \boldsymbol{x}_t} \overline{p}_t(\boldsymbol{x}|\boldsymbol{y}_0, \boldsymbol{\varepsilon}_y) dx. \tag{26}$$

Thus we have

$$\begin{aligned}
& \int e^{ik \cdot \boldsymbol{x}} \partial_t \overline{p}_t(\boldsymbol{x}|\boldsymbol{y}_0, \boldsymbol{\varepsilon}_y) dx \\
&= \int \tilde{\boldsymbol{v}}_t(\boldsymbol{x}_t) \nabla_x (e^{ik \cdot \boldsymbol{x}_t}) \overline{p}_t(\boldsymbol{x}|\boldsymbol{y}_0, \boldsymbol{\varepsilon}_y) dx \\
&= - \int \nabla_x \cdot [\tilde{\boldsymbol{v}}_t(\boldsymbol{x}_t) \overline{p}_t(\boldsymbol{x}|\boldsymbol{y}_0, \boldsymbol{\varepsilon}_y)] e^{ik \cdot \boldsymbol{x}_t} dx,
\end{aligned} \tag{27}$$

where the last equality uses integration by parts. Using the inverse Fourier transform we have

$$\partial_t \overline{p}_t(\boldsymbol{x}|\boldsymbol{y}_0, \boldsymbol{\varepsilon}_y) + \nabla_x \cdot [\tilde{\boldsymbol{v}}_t(\boldsymbol{x}_t) \overline{p}_t(\boldsymbol{x}|\boldsymbol{y}_0, \boldsymbol{\varepsilon}_y)] = 0. \tag{28}$$

Since we start from $t = T$, a minus sign should be added to $\tilde{\boldsymbol{v}}$. Using Assumption 1 gives Eq. (17). Assumption 1 can also be equivalently stated as follows: let the density function of $\boldsymbol{x}_t$ given $\boldsymbol{y}_0$ and $\boldsymbol{\varepsilon}_y$ be $\overline{p}_t(\boldsymbol{x}|\boldsymbol{y}_0, \boldsymbol{\varepsilon}_y)$, then there exists $p_t$ such that

$$\overline{p}_t(\boldsymbol{x}|\boldsymbol{y}_0, \boldsymbol{\varepsilon}_y) = p_t(\boldsymbol{x}|\boldsymbol{y}_t). \tag{29}$$

The equality holds when $\boldsymbol{\varepsilon}_y$ is independent of $\boldsymbol{\varepsilon}_x$. The proof follows the one of Equation 9 by Ma et al. (2024).

$$\begin{aligned}
\boldsymbol{v}_t(\boldsymbol{x}|\boldsymbol{y}_t) &= \dot{\alpha}_t \mathbb{E}\left[\boldsymbol{x}_*|\boldsymbol{x}, \boldsymbol{y}_t\right] + \dot{\sigma}_t \mathbb{E}\left[\boldsymbol{\varepsilon}_x|\boldsymbol{x}, \boldsymbol{y}_t\right] \\
&= \dot{\alpha}_t \mathbb{E}\left[\left.\frac{\boldsymbol{x}_t - \sigma_t \boldsymbol{\varepsilon}}{\alpha_t}\right| \boldsymbol{x}, \boldsymbol{y}_t\right] + \dot{\sigma}_t \mathbb{E}\left[\boldsymbol{\varepsilon}_x|\boldsymbol{x}, \boldsymbol{y}_t\right] \\
&= \frac{\dot{\alpha}_t}{\alpha_t} \boldsymbol{x} + \left(\dot{\sigma}_t - \frac{\dot{\alpha}_t \sigma_t}{\alpha_t}\right) \mathbb{E}\left[\boldsymbol{\varepsilon}_x|\boldsymbol{x}, \boldsymbol{y}_t\right] \\
&= \frac{\dot{\alpha}_t}{\alpha_t} \boldsymbol{x} + \left(\dot{\sigma}_t - \frac{\dot{\alpha}_t \sigma_t}{\alpha_t}\right) (-\sigma_t \boldsymbol{s}_t(\boldsymbol{x}|\boldsymbol{y}_t)) \\
&= \frac{\dot{\alpha}_t}{\alpha_t} \boldsymbol{x} - \lambda_t \sigma_t \boldsymbol{s}_t(\boldsymbol{x}|\boldsymbol{y}_t)
\end{aligned} \tag{30}$$

## A.2 PROOF OF COROLLARY 1

From Eq. (16), the first update for $\boldsymbol{x}'_{i-1}$ uses information of $\boldsymbol{x}_i$. If we now consider the second step first and then it's next update step, the update scheme reads as follows:

$$\begin{cases} \boldsymbol{x}_{i-1} & = \boldsymbol{x}'_{i-1} - \frac{\lambda_t \sigma_t}{2\alpha_t^2 \sigma_n^2} \nabla_{x'_{i-1}} \|\boldsymbol{y}_{i-1} - \boldsymbol{A}\boldsymbol{x}'_{i-1}\|^2 \Delta t \\ \boldsymbol{x}'_{i-2} & = \boldsymbol{x}_{i-1} - \boldsymbol{v}_{t_{i-1}}(\boldsymbol{x}_{i-1})\Delta t, \end{cases} \tag{31}$$

which is a splitting of

$$-\Big(\frac{\dot{\alpha}_{t_{i-1}}}{\alpha_{t_{i-1}}}\boldsymbol{x} - \lambda_{t_{i-1}}\sigma_{t_{i-1}}\nabla_x \log p_{t_{i-1}}(\boldsymbol{x}|\boldsymbol{y}_{t_{i-1}})\Big)dt. \tag{32}$$

This completes the proof.

## A.3 PROOF OF COROLLARY 2

The corollary is a direct result of Theorem 1. The velocity field of diffusion models is defined in the probability flow ODE (Song et al., 2021b), which is exactly Eq. (22) when conditioning on $\boldsymbol{y}_t$.

# B ABLATION STUDY

## B.1 $K$-STEPS CONDITIONAL UPDATE

In this section, we perform the ablation study over the $K$-steps conditional update strategy.

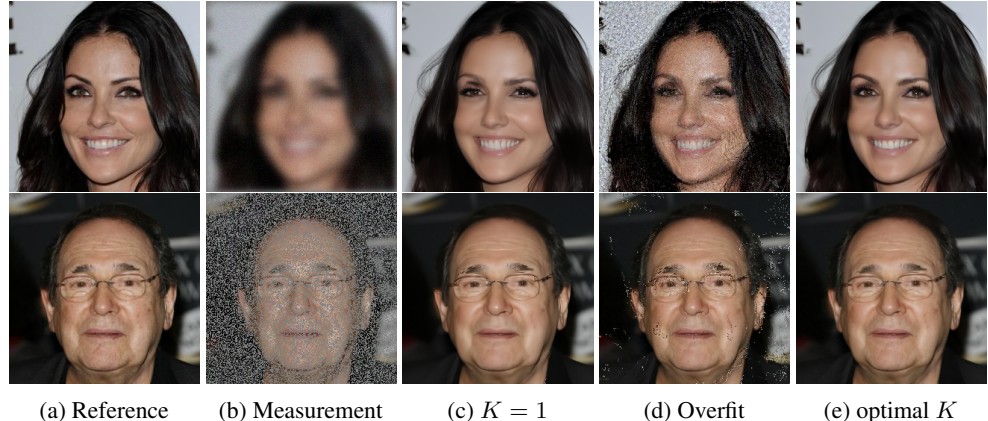

(a) Reference     (b) Measurement     (c) $K = 1$     (d) Overfit     (e) optimal $K$

Figure 7: Our FIG algorithm for Gaussian deblurring and random inpainting with noise $\sigma_n = 0.05$ on the CelebA-HQ dataset.

| Method | SR ($4\times$) | | | Gaussian Deblur | | | Colorization | | | Inpainting | | |
|---|---|---|---|---|---|---|---|---|---|---|---|---|
| | PSNR ↑ | SSIM ↑ | LPIPS ↓ | PSNR ↑ | SSIM ↑ | LPIPS ↓ | PSNR ↑ | SSIM ↑ | LPIPS ↓ | PSNR ↑ | SSIM ↑ | LPIPS ↓ |
| $K = 1$ | 28.51 | 0.812 | 0.222 | 25.43 | 0.691 | 0.271 | 24.21 | 0.902 | 0.241 | 30.02 | 0.895 | 0.189 |
| optimal $K$ | 28.51 | 0.812 | 0.222 | 26.45 | 0.738 | 0.257 | 24.21 | 0.902 | 0.241 | 33.33 | 0.920 | 0.131 |

Table 4: Quantitative comparison (PSNR, SSIM, LPIPS) of our algorithm for different $K$'s on the CelebA-HQ $256 \times 256$ validation set. All input images have a measurement Gaussian noise of $\sigma_n = 0.05$. Note that for SR ($4\times$) and colorization, the optimal $K = 1$, so they have the same metrics.

| Method | SR (4×) | | | Gaussian Deblur | | | Colorization | | | Inpainting | | |
|---|---|---|---|---|---|---|---|---|---|---|---|---|
| | PSNR ↑ | SSIM ↑ | LPIPS ↓ | PSNR ↑ | SSIM ↑ | LPIPS ↓ | PSNR ↑ | SSIM ↑ | LPIPS ↓ | PSNR ↑ | SSIM ↑ | LPIPS ↓ |
| $K = 1$ | 24.48 | 0.646 | 0.370 | 22.75 | 0.533 | 0.445 | 24.27 | 0.862 | 0.284 | 27.25 | 0.815 | 0.246 |
| optimal $K$ | 24.48 | 0.646 | 0.370 | 23.36 | 0.554 | 0.433 | 24.27 | 0.862 | 0.284 | 29.17 | 0.872 | 0.156 |

Table 5: Quantitative comparison (PSNR, SSIM, LPIPS) of our algorithm for different $K$'s on the LSUN-Bedroom $256 \times 256$ validation set. All input images have a measurement Gaussian noise of $\sigma_n = 0.05$. Note that for SR (4×) and colorization, the optimal $K = 1$, so they have the same metrics.

| Method | SR (4×) | | | Gaussian Deblur | | | Colorization | | | Inpainting | | |
|---|---|---|---|---|---|---|---|---|---|---|---|---|
| | PSNR ↑ | SSIM ↑ | LPIPS ↓ | PSNR ↑ | SSIM ↑ | LPIPS ↓ | PSNR ↑ | SSIM ↑ | LPIPS ↓ | PSNR ↑ | SSIM ↑ | LPIPS ↓ |
| $K = 1$ | 28.45 | 0.730 | 0.260 | 25.33 | 0.590 | 0.330 | 25.52 | 0.906 | 0.246 | 31.04 | 0.867 | 0.178 |
| optimal $K$ | 28.45 | 0.730 | 0.260 | 26.25 | 0.632 | 0.310 | 25.52 | 0.906 | 0.246 | 31.04 | 0.867 | 0.178 |

Table 6: Quantitative comparison (PSNR, SSIM, LPIPS) of our algorithm for different $K$'s on the AFHQ-Cat $256 \times 256$ validation set. All input images have a measurement Gaussian noise of $\sigma_n = 0.05$. Note that for SR (4×), colorization, and inpainting, the optimal $K = 1$, so they have the same metrics.

We conduct experiments on a separate validation set (100 images from the official test data split) using different values of $K$ and report results for $K = 1$ and the optimal $K$. In each experiment, we finetune other hyperparameters so that our algorithm achieves the best performance for the given $K$. Fig. 7 contains examples of Gaussian deblurring and random inpainting on the CelebA-HQ dataset. When we set $K = 1$, the results are blurry and exhibit unnatural details compared to the result for the optimal $K$. If we fully minimize the distance $\|\boldsymbol{y}_t - \boldsymbol{A}\boldsymbol{x}_t\|_2^2$ at each timestep, the output shows severe overfitting problems as illustrated in Fig. 7d. To mitigate the risk, we tune $K$ and $c$ for better performance. The quantitative results are shown in Table 4, Table 5, and Table 6.

## B.2 Measurement Interpolant Variance Rescaling

In this section, we ablate the measurement interpolant variance rescaling technique.

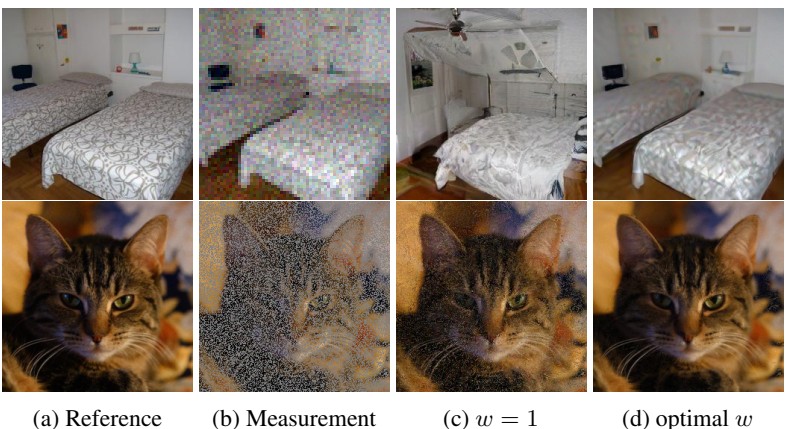

(a) Reference  (b) Measurement  (c) $w = 1$  (d) optimal $w$

Figure 8: Our FIG algorithm for super-resolution and random inpainting with noise $\sigma_n = 0.05$ on the LSUN-Bedroom and AFHQ-Cat datasets.

| Method | SR ($4\times$) | | | Gaussian Deblur | | | Colorization | | | Inpainting | | |
|---|---|---|---|---|---|---|---|---|---|---|---|---|
| | PSNR ↑ | SSIM ↑ | LPIPS ↓ | PSNR ↑ | SSIM ↑ | LPIPS ↓ | PSNR ↑ | SSIM ↑ | LPIPS ↓ | PSNR ↑ | SSIM ↑ | LPIPS ↓ |
| $w = 1$ | 25.66 | 0.585 | 0.376 | 24.08 | 0.533 | 0.383 | 23.24 | 0.673 | 0.400 | 19.36 | 0.407 | 0.495 |
| optimal $w$ | 28.51 | 0.812 | 0.222 | 26.45 | 0.738 | 0.257 | 24.21 | 0.902 | 0.241 | 33.33 | 0.920 | 0.131 |

Table 7: Quantitative comparison (PSNR, SSIM, LPIPS) of our algorithm for different $w$'s on the CelebA-HQ $256 \times 256$ validation set. All input images have a measurement Gaussian noise of $\sigma_n = 0.05$.

| Method | SR ($4\times$) | | | Gaussian Deblur | | | Colorization | | | Inpainting | | |
|---|---|---|---|---|---|---|---|---|---|---|---|---|
| | PSNR ↑ | SSIM ↑ | LPIPS ↓ | PSNR ↑ | SSIM ↑ | LPIPS ↓ | PSNR ↑ | SSIM ↑ | LPIPS ↓ | PSNR ↑ | SSIM ↑ | LPIPS ↓ |
| $w = 1$ | 21.02 | 0.327 | 0.566 | 20.06 | 0.273 | 0.575 | 19.80 | 0.486 | 0.557 | 19.48 | 0.346 | 0.571 |
| optimal $w$ | 24.48 | 0.646 | 0.370 | 23.36 | 0.554 | 0.433 | 24.27 | 0.862 | 0.284 | 29.17 | 0.872 | 0.156 |

Table 8: Quantitative comparison (PSNR, SSIM, LPIPS) of our algorithm for different $w$'s on the LSUN-Bedroom $256 \times 256$ validation set. All input images have a measurement Gaussian noise of $\sigma_n = 0.05$.

| Method | SR ($4\times$) | | | Gaussian Deblur | | | Colorization | | | Inpainting | | |
|---|---|---|---|---|---|---|---|---|---|---|---|---|
| | PSNR ↑ | SSIM ↑ | LPIPS ↓ | PSNR ↑ | SSIM ↑ | LPIPS ↓ | PSNR ↑ | SSIM ↑ | LPIPS ↓ | PSNR ↑ | SSIM ↑ | LPIPS ↓ |
| $w = 1$ | 26.02 | 0.538 | 0.395 | 24.74 | 0.466 | 0.419 | 23.95 | 0.685 | 0.441 | 21.76 | 0.388 | 0.523 |
| optimal $w$ | 28.45 | 0.730 | 0.260 | 26.25 | 0.632 | 0.310 | 25.52 | 0.906 | 0.246 | 31.04 | 0.867 | 0.178 |

Table 9: Quantitative comparison (PSNR, SSIM, LPIPS) of our algorithm for different $w$'s on the AFHQ-Cat $256 \times 256$ validation set. All input images have a measurement Gaussian noise of $\sigma_n = 0.05$.

In our algorithm, the measurement interpolant is defined as $\boldsymbol{y}_t = \alpha_t \boldsymbol{y}_0 + w \sigma_t \boldsymbol{\varepsilon}_y$. Without rescaling, i.e., for $w = 1$, as shown in the first row in Fig. 8, we observe that the reconstructed result appears as if two images have been combined: one representing our measurement and the other determined by the initialization and the ODE. Increasing the weight of the measurement guidance can address this issue, but it leads to overfitting, resulting in a high level of noise in the final output as shown in the second row in Fig. 8. With a smaller optimal $w$, we can achieve a balance between these two situations, resulting in much better outcomes. The quantitative results are presented in Tables 7 to 9. All experiments are conducted on the validation set by carefully fine-tuning the hyper-parameters. Fig. 9 gives a visual illustration of effects on different $w$ for super resolution on CelebA with flow matching prior. It's clear that $w$ affects the performance of our algorithm and there exists an optimal $w$. We only show the PSNR and LPIPS because the SSIM scores are close.

## C  FIG+ ALGORITHM

We provide the FIG+ algorithm in Algorithm 2.

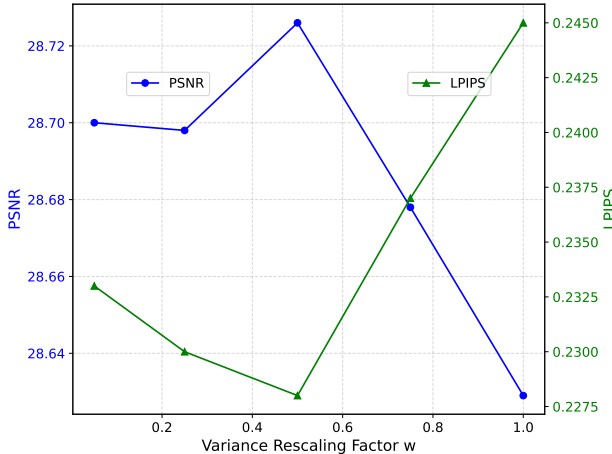

Figure 9: Metrics for $4\times$ super resolution on CelebA-HQ $256 \times 256$ validation set with different variance rescaling factor $w$.

---

**Algorithm 2** FIG+

---

**Require:** $T, c, K, w, m, \boldsymbol{y}_0$
1: Initialize $\boldsymbol{x}_T = \boldsymbol{\varepsilon}_x \sim \mathcal{N}(\boldsymbol{0}, \boldsymbol{I})$             $\triangleright$ Initialize $\boldsymbol{x}_t$
2: $\Delta t = 1/T$
3: **for** $i = T$ **to** $1$ **do**
4:   $t = i/T, t' = (i-1)/T$
5:   $\boldsymbol{y}_{i-1} = \alpha_{t'}\boldsymbol{y}_0 + w\sigma_{t'}\boldsymbol{A}\boldsymbol{\varepsilon}_x$      $\triangleright$ measurement interpolation with rescaled variance
6:   $\boldsymbol{x}_{i-1} = \boldsymbol{x}_i - \boldsymbol{v}_\theta(\boldsymbol{x}_i, t)\Delta t$           $\triangleright$ Unconditional update
7:   **for** $k = 1$ **to** $K$ **do**
8:    $\boldsymbol{x}_{i-1} = \boldsymbol{x}_{i-1} - \frac{c\lambda_t \sigma_t \Delta t}{2\alpha_t^2 \sigma_n^2}\nabla_{\boldsymbol{x}_{i-1}}\|\boldsymbol{y}_{i-1} - \boldsymbol{A}\boldsymbol{x}_{i-1}\|_2^2$    $\triangleright$ $K$-steps conditional update
9:   **end for**
10:   $\boldsymbol{x}_0' = (1-t)\boldsymbol{v}_\theta(\boldsymbol{x}_i, t) + \boldsymbol{x}_i$       $\triangleright$ use Tweedie's formula to estimate $\boldsymbol{x}_0$
11:   $\boldsymbol{x}_{i-1}' = \alpha_{i-1}\boldsymbol{x}_0' + \sigma_{i-1}\boldsymbol{\varepsilon}_x$        $\triangleright$ perturb estimated $\boldsymbol{x}_0$ to time $t_{i-1}$
12:   $\boldsymbol{x}_{i-1} = \boldsymbol{A}\boldsymbol{x}_{i-1} + (1-m)(\boldsymbol{I} - \boldsymbol{A})\boldsymbol{x}_{i-1} + m(\boldsymbol{I} - \boldsymbol{A})\boldsymbol{x}_{i-1}'$    $\triangleright$ mix $\boldsymbol{x}_{i-1}'$ with $\boldsymbol{x}_{i-1}$
13: **end for**

---

## D   RECOVERY VS PERCEPTUAL TRADE-OFF

| **Method** | PSNR $\uparrow$ | SSIM $\uparrow$ | LPIPS $\downarrow$ | FID $\downarrow$ | KID $\downarrow$ |
|---|---|---|---|---|---|
| FIG-Diffusion | **29.89** | **0.846** | 0.163 | 25.46 | 0.013 |
| DPS-Diffusion | 28.85 | 0.801 | 0.172 | **19.31** | **0.005** |
| DDNM+ | 28.82 | 0.750 | 0.340 | 64.85 | 0.052 |
| DAPS | 29.59 | 0.809 | 0.158 | 23.69 | 0.013 |
| C-$\Pi$GDM | 28.14 | 0.787 | **0.106** | 30.51 | 0.019 |

Table 10: Quantitative comparison (PSNR, SSIM, LPIPS, FID, KID) of different algorithms for $4\times$ super-resolution with noise $\sigma_n = 0.05$ on the CelebA-HQ $256 \times 256$ test dataset. **Bold** for the best.

In this section, we report FID and KID along with the metrics used in the main paper, i.e., PSNR, SSIM, and LPIPS for $4\times$ super-resolution with noise $\sigma_n = 0.05$ on the CelebA-HQ $256 \times 256$ (see Table 10), and discuss the recovery and perceptual trade-off in those metrics.

PSNR and SSIM are considered recovery metrics that measure pixel-level fidelity while LPIPS, FID, and KID are referred to as perceptual metrics that assess perceptual similarity or high-level semantic fidelity.

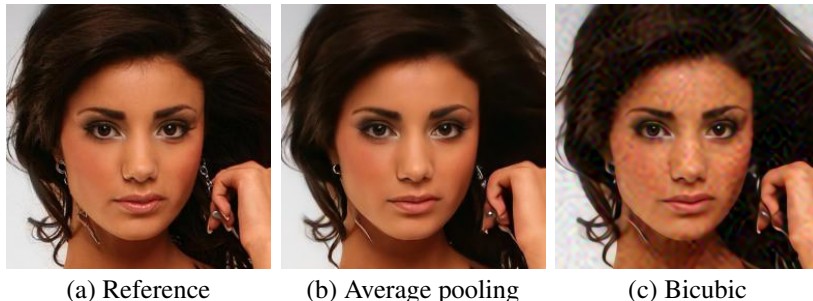

| (a) Reference | (b) Average pooling | (c) Bicubic |

Figure 10: DDNM+ results for $4\times$ bicubic super-resolution with noise $\sigma_n = 0.05$ on the CelebA-HQ dataset.

In this paper, we focus on image reconstruction tasks involving noisy measurements. In this case, metrics like FID and KID, while effective for generative models emphasizing perceptual realism, evaluate distribution-level similarity and may fail to capture details in structural recovery, particularly in high-noise scenarios where subtle details matter. Additionally, FID and KID can be misleading if the reconstructed images are perceptually plausible but deviate significantly from the ground truth. PSNR and SSIM, however, provide a clear and objective measure of how well the noise has been suppressed and the original content preserved, which remains critical in our experiments. In all, PSNR and SSIM, LPIPS are baseline metrics widely used in image restoration domain.

As shown in Table 10, our algorithm achieves a balanced trade-off between recovery and perceptual metrics. In the task of noisy image reconstruction, it not only delivers the best recovery metrics but also achieves impressive perceptual scores.

## E  BASELINE IMPLEMENTATIONS

For the pretrained model, we use both flow matching and diffusion models. For all algorithms with a flow matching base model, we use the rectified flow.[1] For measurement operators, since our baseline methods OT-ODE and DMPS require an SVD, we inherit codes from the DDRM Github repository.

**DPS-Flow.**  In the codes for the Rectified flow model, $\alpha_t = t$ and $\sigma_t = 1 - t$. Following Eq. (18), we have $\hat{x}_0 = (1 - t)v_\theta(x_t, t) + x_t$. Given $\hat{x}_0$, we implement DPS strictly following Algorithm 1 of Chung et al. (2023b) and fine-tune the hyperparameter $\zeta_i$ to achieve the best performance.

**DMPS.**  With the SVD operations from DDRM codes, we implement DMPS strictly following Algorithm 1 in Meng & Kabashima (2022) and fine-tune the hyperparameter $\lambda$ to get the best results. Again, the score function is obtained by the velocity following Eq. (18).

**OT-ODE.**  Using the same $\hat{x}_0$ as DPS and the SVD operations from the DDRM implementation, we implement the OT-ODE strictly following Algorithm 1 of Pokle et al. (2024) and test the start time for each task to get the best performance.

**DAPS.**  We use the official repository[2] and strictly follow the Algorithm 1 of Zhang et al. (2024) to implement DAPS.

**DPS-Diffusion.**  We implement DPS on the EDM implementation from the DAPS repository.

**DDNM/DDNM+.**  We implement DDNM and DDNM+ strictly following Algorithm 1 and 2 of Wang et al. (2023).[3] We reproduce all the metrics of the tasks reported in their paper but find that the performance of DDNM/DDNM+ varies significantly across different tasks. Our experiment shows that DDNM/DDNM+ fails on Gaussian deblurring with noise, which matches the result of Meng & Kabashima (2022). We also find that DDNM/DDNM+ can't handle super-resolution with bicubic down-sampling well, resulting in noisy reconstructions as shown in Fig. 10. However for super-resolution with average pooling, we achieved good results for both SR($4\times$) with noise standard

---

[1]https://github.com/gnobitab/RectifiedFlow
[2]https://github.com/zhangbingliang2019/DAPS
[3]https://github.com/wyhuai/DDNM

| Method | SR ($4\times$) | | | Gaussian Deblur | | | Colorization | | | Inpainting | | |
|---|---|---|---|---|---|---|---|---|---|---|---|---|
| | PSNR ↑ | SSIM ↑ | LPIPS ↓ | PSNR ↑ | SSIM ↑ | LPIPS ↓ | PSNR ↑ | SSIM ↑ | LPIPS ↓ | PSNR ↑ | SSIM ↑ | LPIPS ↓ |
| FIG (ours) | **24.54** | **0.682** | **0.327** | **22.94** | **0.588** | 0.402 | 21.87 | **0.865** | **0.320** | **30.24** | **0.895** | **0.139** |
| DPS | 24.23 | 0.649 | 0.346 | 19.82 | 0.433 | 0.471 | 14.30 | 0.610 | 0.596 | 27.21 | 0.771 | 0.283 |
| DMPS | 24.08 | 0.653 | 0.368 | 22.76 | 0.575 | 0.413 | 21.26 | **0.865** | 0.326 | 27.34 | 0.841 | 0.212 |
| OT-ODE | 23.82 | 0.639 | 0.349 | 22.16 | 0.566 | **0.389** | **22.06** | 0.781 | 0.385 | 26.56 | 0.768 | 0.278 |

Table 11: Quantitative comparison (PSNR, SSIM, LPIPS) of different algorithms for different tasks on the LSUN-Bedroom $256 \times 256$ test dataset. All input images have a measurement Gaussian noise of $\sigma_n = 0.05$. **Bold** for the best.

| Method | SR ($4\times$) | | | Gaussian Deblur | | | Colorization | | | Inpainting | | |
|---|---|---|---|---|---|---|---|---|---|---|---|---|
| | PSNR ↑ | SSIM ↑ | LPIPS ↓ | PSNR ↑ | SSIM ↑ | LPIPS ↓ | PSNR ↑ | SSIM ↑ | LPIPS ↓ | PSNR ↑ | SSIM ↑ | LPIPS ↓ |
| FIG (ours) | **27.50** | 0.707 | **0.272** | **25.14** | **0.591** | **0.334** | **26.36** | **0.915** | 0.227 | **28.99** | **0.835** | **0.202** |
| DPS | 25.70 | 0.607 | 0.332 | 21.40 | 0.407 | 0.429 | 21.82 | 0.763 | 0.407 | 26.20 | 0.709 | 0.300 |
| DMPS | 27.49 | **0.709** | 0.286 | 24.97 | 0.577 | 0.343 | 26.12 | 0.913 | **0.221** | 27.02 | 0.781 | 0.262 |
| OT-ODE | 25.95 | 0.617 | 0.326 | 24.42 | 0.536 | 0.357 | 25.39 | 0.849 | 0.291 | 28.22 | 0.789 | 0.235 |

Table 12: Quantitative comparison (PSNR, SSIM, LPIPS) of different algorithms for different tasks on the AFHQ-Cat $256 \times 256$ test dataset. All input images have a measurement Gaussian noise of $\sigma_n = 0.05$. **Bold** for the best.

deviation $0.05$ and SR($16\times$) with noise standard deviation $0.2$. Motion deblur is not implemented since DDNM requires SVD operations.

# F ADDITIONAL EXPERIMENTAL RESULTS FOR FLOW MATCHING MODEL

We show additional results with flow matching priors. Reconstructed images for motion deblurring with $\sigma_n = 0.05$ on CelebA-HQ are shown in Fig. 11. Additional quantitative results on other datasets (LSUN-Bedroom, AFHQ-Cat) are shown in Tables 11 and 12.

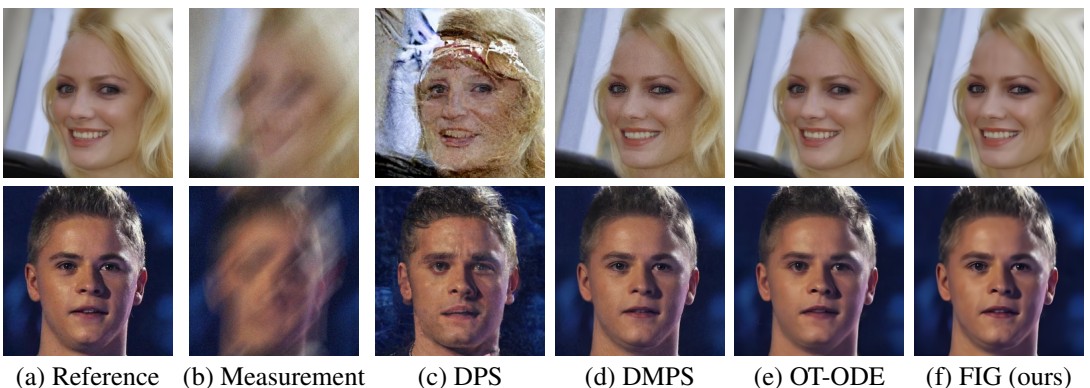

   (a) Reference   (b) Measurement      (c) DPS       (d) DMPS     (e) OT-ODE   (f) FIG (ours)

Figure 11: Results for motion deblurring with noise $\sigma_n = 0.05$ on the CelebA-HQ dataset.

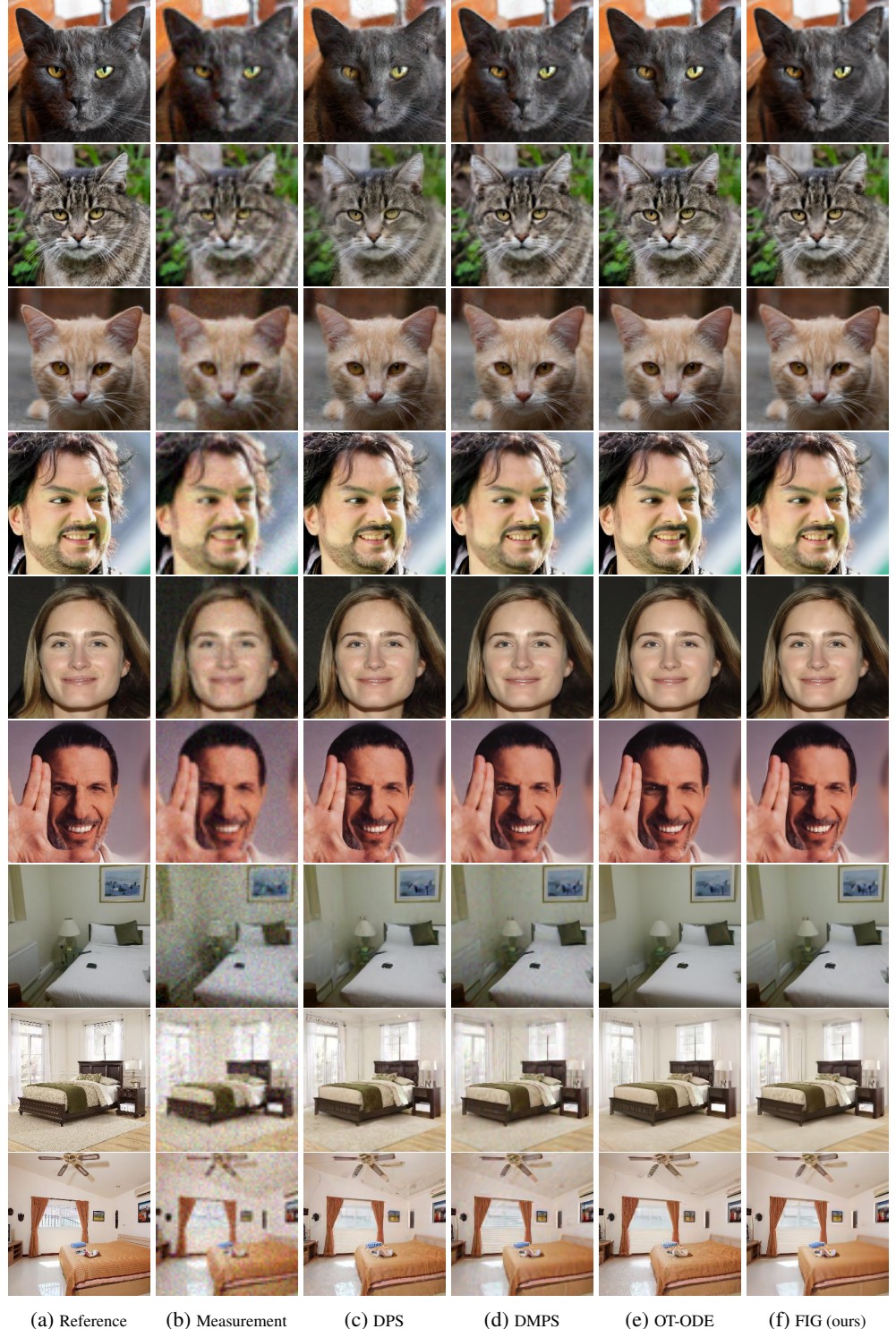

(a) Reference    (b) Measurement    (c) DPS    (d) DMPS    (e) OT-ODE    (f) FIG (ours)

Figure 12: Results for super-resolution ($4\times$) with noise $\sigma_n = 0.05$ on AFHQ-Cat, CelebA-HQ, and LSUN-Bedroom datasets.

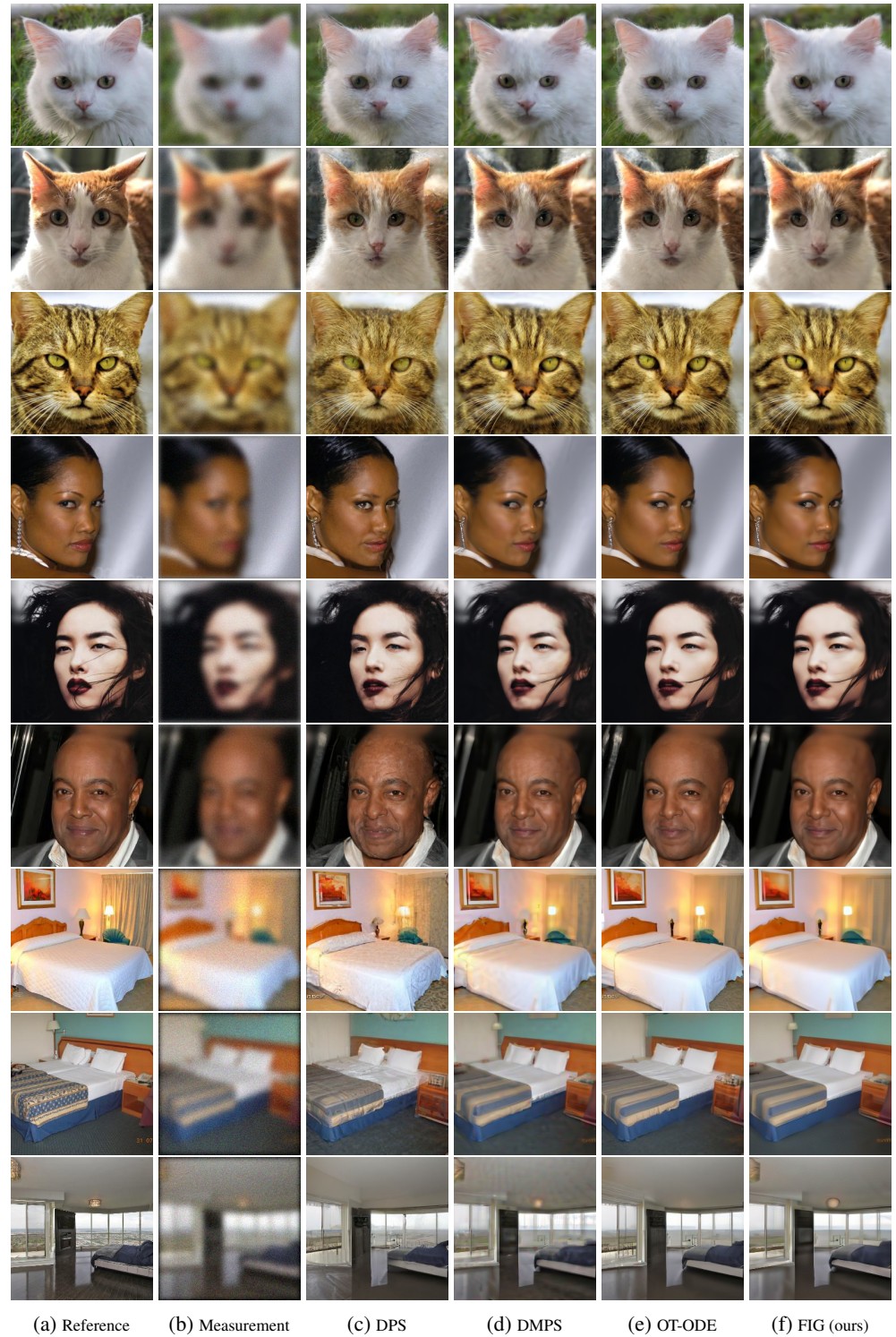

(a) Reference     (b) Measurement     (c) DPS     (d) DMPS     (e) OT-ODE     (f) FIG (ours)

Figure 13: Results for Gaussian deblurring with noise $\sigma_n = 0.05$ on AFHQ-Cat, CelebA-HQ, and LSUN-Bedroom datasets.

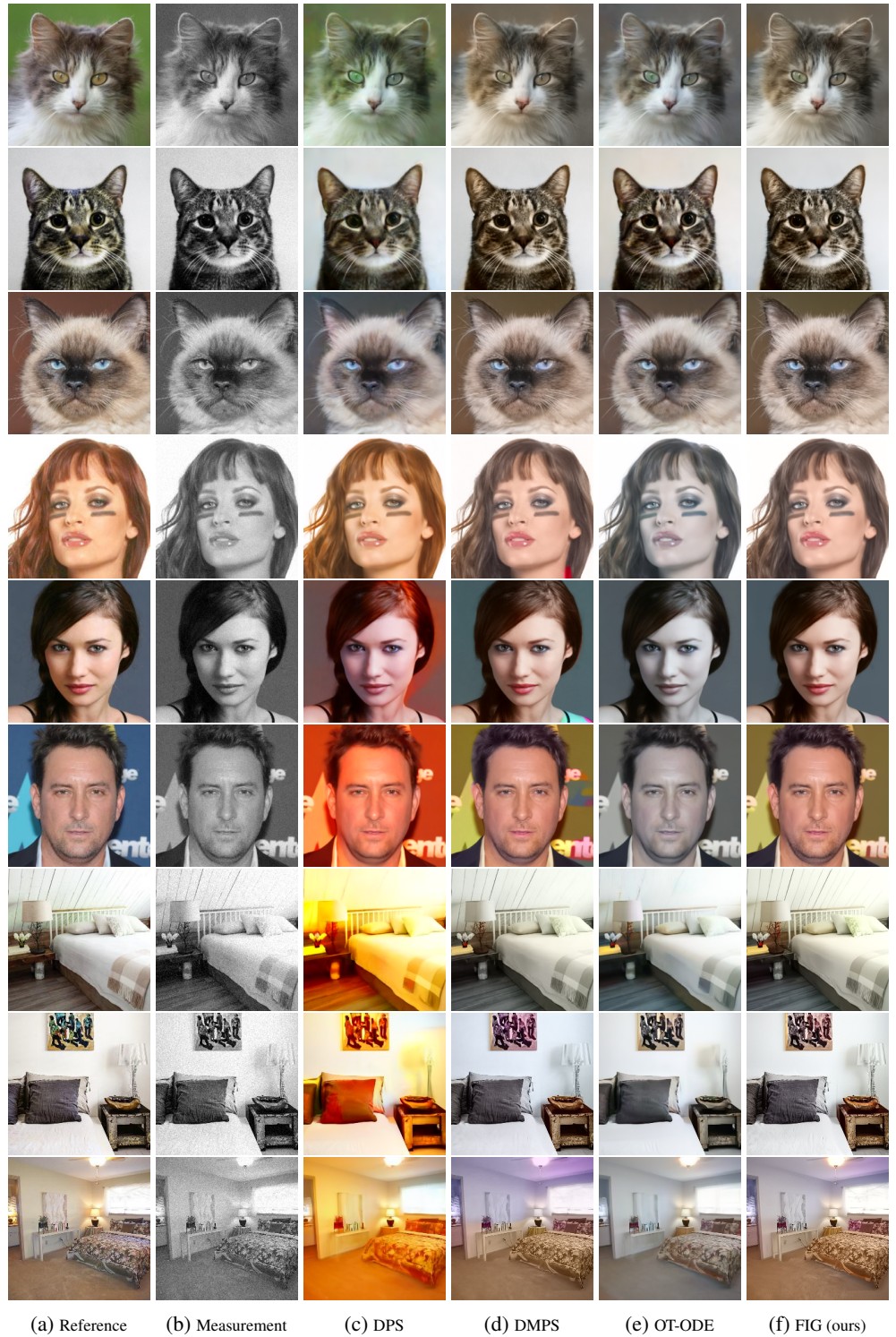

(a) Reference  (b) Measurement  (c) DPS  (d) DMPS  (e) OT-ODE  (f) FIG (ours)

Figure 14: Results for colorization with noise $\sigma_n = 0.05$ on AFHQ-Cat, CelebA-HQ, and LSUN-Bedroom datasets.

# G  ADDITIONAL EXPERIMENT RESULTS FOR DIFFUSION MODEL

In this section, we show that our algorithm works for diffusion models. Qualitative results are shown below.

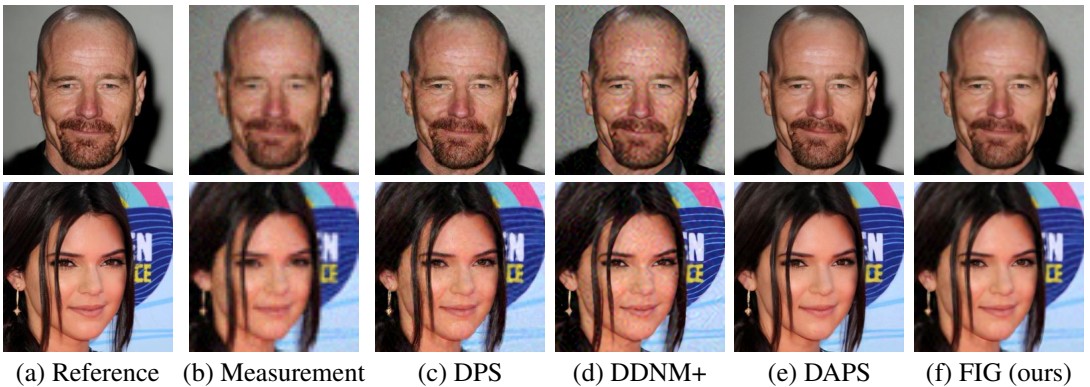

   (a) Reference    (b) Measurement    (c) DPS    (d) DDNM+    (e) DAPS    (f) FIG (ours)

Figure 15: Results for $4\times$ bicubic super-resolution with noise $\sigma_n = 0.05$ on the CelebA-HQ dataset.

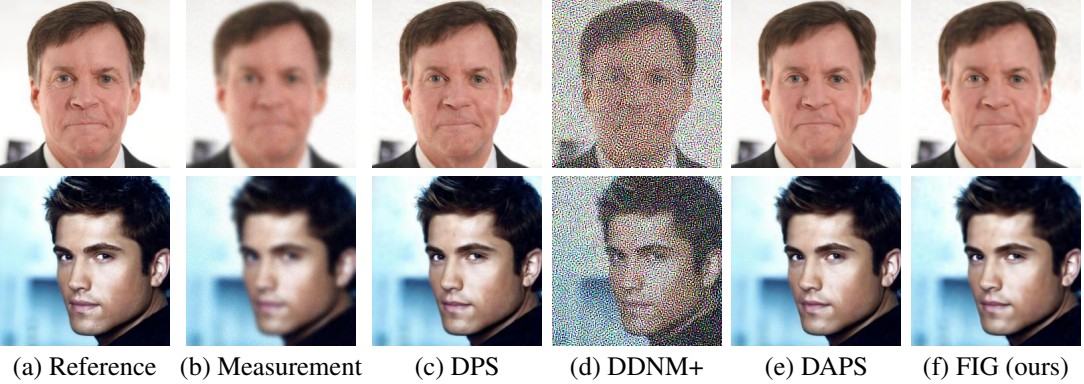

   (a) Reference    (b) Measurement    (c) DPS    (d) DDNM+    (e) DAPS    (f) FIG (ours)

Figure 16: Results for Gaussian deblurring with noise $\sigma_n = 0.05$ on the CelebA-HQ dataset.

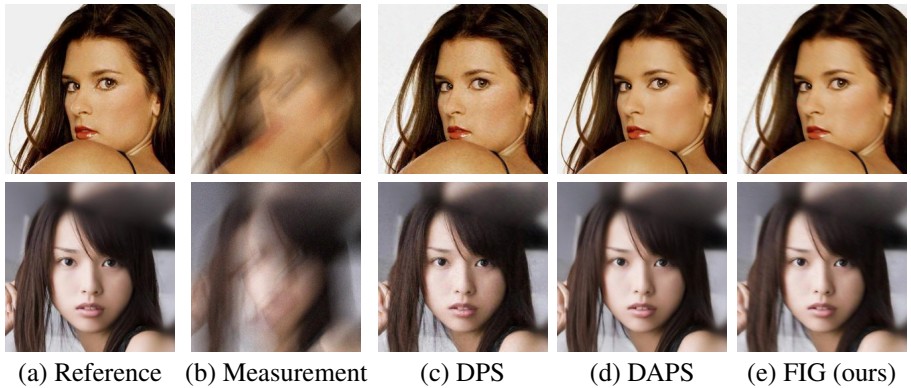

   (a) Reference    (b) Measurement    (c) DPS    (d) DAPS    (e) FIG (ours)

Figure 17: Results for motion deblurring with noise $\sigma_n = 0.05$ on CelebA-HQ dataset.

