# OpenReview forum: "FIG: Flow with Interpolant Guidance for Linear Inverse Problems"
_ICLR.cc/2025/Conference — ICLR 2025 Poster_

### Official Review · Reviewer_iHSi · 2024-10-20

**Soundness:** 3
**Presentation:** 3
**Contribution:** 2
**Rating:** 6
**Confidence:** 2

**Summary:**

This paper introduces an algorithm that efficiently guides reverse-time sampling using measurement interpolants, supported by theoretically grounded frameworks, to tackle image restoration tasks. The performance in the experiment is strong and surpasses previous methods like DDNM+. However, the novelty of the guidance design appears somewhat limited, and the scope of experimental comparisons could be expanded to provide a more convincing evaluation.

**Strengths:**

1. The proposed method is sound and the theoretical framework is well-written with mathematical formulations.
2. The experimental results seem nice.
3. The writing is clear and concise.

**Weaknesses:**

1. The method of using the gradient of reconstruction function to guide the generation direction has been widely explored in recent works. The technique in FIG lacks sufficient novelty in comparison to prior works.
2. The experiments can be more detailed. The author can involve more related works like GDP[a] in comparisons.
3. The effectiveness of the measurement interpolation is not verified in the experiments. The author should analyze it with some ablation studies.
4. The method has not been evaluated in blind scenarios. All degradation settings in this paper are fixed and known. However, real-world images suffer from complex and blind degradations. The author should show the performance on real-world samples.

[a] Generative Diffusion Prior for Unified Image Restoration and Enhancement, CVPR 2023.

**Questions:**

Performance with more base models. While the current experimental setup effectively demonstrates the validity of your approach, it would be interesting to assess how it performs with more recent open-source diffusion and flow models, like Flux 1.0, SD3, SDXL and other DiTs.

---

> ### Author Response · Authors · 2024-11-21
>
> We thank reviewer iHSi for the valuable feedback. We address your concerns and questions below.
>
> ***QD1: Novelty***
> > We proposed a new measurement interpolant guided scheme for solving linear inverse problems using diffusion and flow based prior models. Different from the existing gradient-based algorithms such as DPS, DDNM, OT-ODE, and $\Pi$GDM, our approach avoids the time-consuming backpropagation through the score or flow function. We also would like to point out that the linear inverse problems with extremely noisy measurements is still very challenging for existing algorithms. For example, in Section 4, we included experiments with
> > - 4x super resolution with high-noise $\sigma_n = 1$, comparing to the commonly used $\sigma_n = 0.05$
> > - 16x super resolution with noise $\sigma_n = 0.2$.
> >
> > Our algorithm with flow matching prior model outperforms the baselines on those challenging tasks (see Table 2, Figure 4, and Figure 5).
>
> ***QD2: Experiments***
> > We thank the reviewer for pointing out the paper GDP to us. However, we choose not to include it as our benchmark due to two reasons. First, GDP has slightly worse results than DDNM in terms of evaluation metrics on super resolution$\times$4 and colorization on ImageNet. As we have already used DDNM for our benchmark, we don’t think it is necessary to compare it with GDP. Second, GDP focuses on noiseless measurements. As our paper investigates noisy cases, GDP does not serve as an appropriate benchmark.
>
> > We added a discussion on GDP in Section 2.3. GDP considered blind image reconstruction tasks, which is an important future extension of FIG. We highlighted this in Section 5 and cited the paper.
>
> ***QD3: Effectiveness of the measurement interpolation***
> > We believe that the effectiveness of our method compared to various benchmarks is fully demonstrated in Table 1 and 2. We show that FIG achieves SOTA among different tasks and noise levels. As the measurement interpolants are incorporated via gradient descent, its effectiveness is revealed in an ablation study on gradient descent steps $K$ in Appendix C.1. Please let us know if there are other ablation studies we can carry out to better address your concerns.
>
>
> ***QD4: Blind scenario***
> > Thanks for pointing out this interesting research direction. Consider the task of blind motion deblurring, where we have diffusion or flow matching priors on both the image and the blurring kernel. When fixing either the kernel or the image, the noise model is linear, thus measurement interpolants can be constructed. We believe using FIG’s gradient descent updating scheme consecutively on the image and kernel would lead to solving the problem. However, we think that this extension is not trivial and is beyond the scope of this paper, as the title indicates our focus on linear inverse problems. We highlight this as an intriguing direction for future research in Section 5.
>
> ***QD5: Other base models***
> > Thanks for the suggestion. Currently, we have not conducted experiments on latent models, as base models are not the primary concern of our paper. In fact, to the best of our knowledge, none of the related works could be directly applied to latent diffusion or flow matching models. They all need to be redesigned specifically to accommodate the encoder-decoder structure and latent space. We agree that latent models present a promising direction, and we are eager to explore the application of our algorithm in this area in future works.

---

> ### Author Response · Authors · 2024-11-26
>
> Thanks a lot for your time and valuable feedback. We hope that our response and the updated paper answered your questions. Please let us know if you have any further questions or comments. Thank you once again, and we look forward to your response.

---

### Official Review · Reviewer_9aCd · 2024-11-01

**Soundness:** 3
**Presentation:** 3
**Contribution:** 3
**Rating:** 6
**Confidence:** 4

**Summary:**

The paper introduces FIG, a novel algorithm that leverages pre-trained flow matching and diffusion models to solve linear inverse problems. The main contributions include:
FIG introduces a simple but effective method of guiding the reverse-time sampling process with measurement interpolants.
The authors provide a detailed theoretical explanation of FIG, deriving the updating scheme and showing its relation to the posterior log-likelihood.
FIG is evaluated on several image reconstruction tasks using flow matching and diffusion models, demonstrating its effectiveness and efficiency compared to state-of-the-art baselines.

**Strengths:**

FIG is a novel approach to solving linear inverse problems that combines pre-trained generative models with a theoretically justified interpolation-based guidance scheme. It distinguishes itself from existing methods by directly incorporating measurement information during the reverse sampling process.
The paper presents a well-defined algorithm with strong theoretical justification. The empirical evaluation is comprehensive, covering various tasks and datasets, with thorough comparisons to relevant baselines.
The paper is clearly written and well-structured. The authors effectively explain the core concepts of flow matching and diffusion models, and provide a detailed explanation of FIG. The algorithm is presented in a clear pseudocode, making it easy to follow.
FIG offers a promising approach for tackling linear inverse problems with pre-trained generative models. Its efficiency, effectiveness, and theoretical grounding suggest a significant contribution to the field.

**Weaknesses:**

The paper acknowledges that FIG, as currently formulated, is not directly applicable to non-linear inverse problems or models with latent encodings. While this is a valid limitation, the authors could further discuss potential extensions or modifications to address these limitations.
The ablation study on the measurement interpolant variance rescaling technique could be further expanded. For example, visualizing the effects of different weight values (w) on the reconstructions could provide a more intuitive understanding of the impact of this parameter.

**Questions:**

To further strengthen the evaluation and demonstrate the broad applicability of FIG, could the authors please consider comparing their approach against additional methods within the same domain, if available.
Could the authors discuss potential strategies for extending FIG to handle non-linear inverse problems?
What modifications would be needed to apply FIG to models with latent encodings?
The paper mentions the possibility of overfitting when applying FIG. Could the authors elaborate on this aspect, providing insights into how to mitigate this risk?

---

> ### Author Response · Authors · 2024-11-21
>
> We thank reviewer 9aCd for the valuable feedback. We address your concerns and questions below.
>
> ***QC1: Potential extensions***
> > Thanks for the suggestions. Our motivation of FIG comes from the observation that, although various algorithms are proposed to solve non-linear inverse problems with potential extension to latent diffusion or flow matching models, none of them could efficiently handle linear image restoration tasks, especially under high noise. We deem the noisy linear case an important problem to be solved before considering more complicated scenarios. To this end we propose FIG that focuses on solving linear inverse problems.
>
> > As we mention in Section 5, our algorithm does not directly extend to non-linear measurements or latent encoding. The reason is that we require the likelihood of the measurement interpolant $y_t$ given the diffusion or flow matching prior $x_t$, which is intractable for general non-linear measurements and latent encoding. However, there is a class of nonlinear inverse problems that FIG could potentially be applied to, namely the bilinear problems such as blind deconvolution. Consider the task of blind motion deblurring [1], where we have diffusion or flow matching priors on both the image and the blurring kernel. We believe using FIG’s gradient descent updating scheme consecutively on the image and kernel would lead to solving the problem. However, we think that this extension is not trivial and is beyond the scope of this paper, as the title indicates our focus on linear inverse problems. We highlight this as an intriguing direction for future research in the updated Section 5.
>
> Reference:
> [1] Chung, Hyungjin, et al. “Parallel diffusion models of operator and image for blind inverse problems. IEEE.” In CVPR, 2023.
>
>
> ***QC2: Additional ablation studies on rescaling***
> > We have added Figure 9 for a visualization of the effect of $w$ in Appendix C. Fig. 9 gives a visual illustration of effects on different $w$ for super resolution on CelebA with flow matching prior. It’s clear that $w$ affects the performance of our algorithm and there exists an optimal $w$.
>
> ***QC3: Additional comparison***
> > We have added a new baseline C-$\Pi$GDM under the recommendation of Reviewer V1uU. Please see the response to QB for more details. In our implementation, we successfully reproduced results in the C-$\Pi$GDM paper for noiseless measurements. The noisy measurement results indicate that C-$\Pi$GDM, although being highly efficient, struggles to deal with measurement noise.
>
>
> ***QC4: The paper mentions the possibility of overfitting when applying FIG. Could the authors elaborate on this aspect, providing insights into how to mitigate this risk?***
> > Thanks for pointing this out. By overfitting, we refer to situations when the algorithm learns the pattern in the noisy measurement, see Fig. 7d. We find that by tuning the gradient descent steps $K$ and the learning rate $c$, overfitting can be easily avoided. Please see Appendix C.1 for intuitive figure illustrations.

---

> ### Author Response · Authors · 2024-11-26
>
> Thanks a lot for your time and valuable feedback. We hope that our response and the updated paper answered your questions. Please let us know if you have any further questions or comments. Thank you once again, and we look forward to your response.

---

### Official Review · Reviewer_V1uU · 2024-11-02

**Soundness:** 2
**Presentation:** 3
**Contribution:** 2
**Rating:** 6
**Confidence:** 4

**Summary:**

This paper proposed a solution for solving linear inverse problem with flow matching model. The author provides both theoritecal justification and experiment results to support the claim that the proposed method can efficiently handle challenging linear inverse problems.

**Strengths:**

This paper is well-written with both theoretical claim and experiment results.
The model can be deployed in both diffusion and flow matching scenarios.

**Weaknesses:**

Technically, the paper lacks significant contributions that clearly distinguish it from prior work. My concerns are as follows:

1. **Inverse Problem Approach**: Solving an inverse problem with pre-trained diffusion/flow matching models typically involves classifier guidance by designing a robust $q(y|x_t)$ term, as Bayes’ rule suggests: $\nabla p(x|y) = \nabla p(x) + \nabla p(y|x)$. However, the authors do not propose a novel method for posterior estimation. Equation (16) indicates that this work adheres to the scheme outlined in the [DPS](https://arxiv.org/pdf/2209.14687) paper, using reconstruction guidance.

2. **Application of Existing Methods**: This paper seems primarily focused on adapting DPS to flow matching models, with only minor adjustments to weighting terms.

3. **Theorem 1**: This theorem has been discussed in previous work. For example, the [stochastic interpolant](https://arxiv.org/pdf/2209.15571) paper (Appendix F) provides a proof connecting unconditional flow and score-based models. Similarly, [OT-ODE](https://arxiv.org/pdf/2310.04432) and [C-PG(D/F)M](https://arxiv.org/pdf/2405.17673) offer similar proofs for inverse problem solutions. Thus, Theorem 1 could be moved to the appendix, rather than highlighted as a primary contribution.

4. **Omission of Related Work**: The paper omits mention of highly related work, such as [C-PG(D/F)M](https://arxiv.org/pdf/2405.17673), which specifically addresses efficiency in solving inverse problems using both flow matching and diffusion models. As this work was posted on arXiv in May, it should be acknowledged, and I recommend that the authors review it thoroughly.

5. **Efficiency Claims**: The authors assert, "Despite promising experimental results, we find prior works struggle to efficiently handle challenging linear inverse problems," which is questionable. It seems the key contribution here is an additional iterative update at each ODE solving step (Algorithm 1, lines 7-8), introducing a hyperparameter $K$. For $K > 1$, this approach could increase inference time due to additional gradient calculations. Although this gradient term may not be computationally prohibitive for linear transformations $A$, it could become costly for non-linear transformations. While the authors focus on the linear case, the method could potentially be extended to non-linear cases using heuristics, as seen in [$\Pi$GDM](https://openreview.net/pdf?id=9_gsMA8MRKQ).

    - **Efficient Inverse Problem Solutions**: Achieving efficiency typically involves (1) reducing NFE (Number of Function Evaluations) without sacrificing quality (e.g., [C-PG(D/F)M](https://arxiv.org/pdf/2405.17673) achieves NFE=5 for inverse problem solutions), and (2) avoiding costly gradient calculations (e.g., [Red-Diff](https://arxiv.org/pdf/2305.04391)).

6. **Evaluation Metrics**: The evaluation could be improved. PSNR and SSIM are no longer reliable metrics for generative model assessment, especially when leveraging pre-trained unconditional models. Instead, metrics like FID and KID should be considered.

**Questions:**

See weakness above

---

> ### Author Response · Authors · 2024-11-21
>
> We thank reviewer V1uU for the valuable feedback. We address your concerns and questions below.
>
> ***QB1: Novelty and Difference from DPS.***
> > We kindly disagree with the reviewer’s assessment that “this work adheres to the scheme outlined in the DPS paper.” We would like to emphasize that our method FIG is completely different from DPS. At $x_t$, DPS uses Tweedie’s formula to estimate $\hat{x}_0(x_t)$, and differentiate the loss $\Vert y - A\hat{x}_0(x_t)\Vert_2$ to update $x_t$. This involves differentiating through the score or velocity network due to the Tweedie term $\hat{x}_0(x_t)$. In FIG, we use the forward process to interpolate the measurement $y$ to measurement interpolants $y_t$, which is detailed in Section 3.1. We then use the loss $\Vert y_t - Ax_t\Vert_2$ to update $x_t$, which does not require differentiating through the score/velocity network. Equation (16) makes our point self-evident. Furthermore, FIG and DPS yield different reconstruction images, evaluation scores, and runtime efficiency as shown in Section 4. This is clear evidence that FIG is different from DPS.
>
> > In FIG, we proposed a new measurement interpolant guided scheme for solving linear inverse problems using diffusion and flow based prior models. We also would like to point out that the linear inverse problems with extremely noisy measurements is still very challenging for existing algorithms. For example, in Section 4, we included experiments with
> 4x super resolution with high-noise $\sigma_n = 1$, comparing to the commonly used $\sigma_n = 0.05$
> 16x super resolution with noise $\sigma_n = 0.2$.
> Our algorithm with flow matching prior model outperforms the existing algorithm on those challenging tasks (see Table 2, Figure 4, and Figure 5).
>
> ***QB2: This paper seems primarily focused on adapting DPS to flow matching models.***
> > As is explained in QB1, FIG is different from DPS. Moreover, we implemented DPS with flow matching models as a baseline for comparison. Appendix E details how we implement DPS-Flow.
>
> ***QB3: Theorem 1 has been discussed in previous work.***
> > We have acknowledged previous works that provide similar proofs in Appendix A.1 in the original paper. However, since FIG’s conditioning is different from previous methods, we believe that showing the dynamics (Theorem 1) of FIG in the main paper is important. As mentioned in Section 3.3, the FIG updating scheme is a numerical discretization of Eq. (20) that corresponds to the dynamical property in Eq. (17), thus suggesting the importance of its justification in Theorem 1. We believe that delaying Theorem 1 to the Appendix will affect the completeness of the main paper.
>
> ***QB4: The paper omits mention of highly related work, such as C-PG(D/F)M.***
> > Thanks for pointing out. We have added the C-$\Pi$GDM baseline to the experiments (Section 4) and related works (Section 2.3). We are aware of the fact that C-$\Pi$GDM includes a flow matching version. Yet due to the fact that only the diffusion version is released in the corresponding Git repo, we added the diffusion version to our baseline. In our implementation, we successfully reproduced results in the C-$\Pi$GDM paper for noiseless measurements. The noisy measurement results indicate that C-$\Pi$GDM, although being highly efficient, struggles to deal with measurement noise.
>
> ***QB5: Efficiency***
> > As is explained in the answer of QB1, FIG is efficient as it does not need to differentiate through the pre-trained neural net. Table 3 empirically demonstrates FIG’s efficiency. We acknowledge that C-$\Pi$GDM is, to the best of our knowledge, the most efficient algorithm for image inverse problems with diffusion or flow matching prior. They cleverly adopt a strategy to warm start from $t=0.4T$ with specific initializations, substantially reducing the NFEs. Empirically, we observe that although C-$\Pi$GDM is very fast, their reconstruction qualities are worse than other baseline algorithms for Gaussian deblurring and Inpainting with noisy measurements (see the updated Table 1).
>
> ***QB6: Evaluation Metrics.***
> > We reported the FID and KIDs and discussed the recovery-perceptual trade-off in the newly added Appendix D. In noisy image reconstruction tasks, dismissing PSNR and SSIM undervalues their critical role in assessing reconstruction fidelity. These metrics ensure pixel-level accuracy, essential for recovering precise details from degraded inputs. While FID and KID capture perceptual realism, they may miss structural details in high-noise scenarios or misrepresent results if reconstructed images deviate from the ground truth since they measure distance in feature space. Also, from the reported perceptual metrics in the newly added Table 10, we observe that our algorithm delivers the best recovery metrics and also achieves comparable perceptual scores.

---

> > ### Comment · Reviewer_V1uU · 2024-11-21
> > **Thanks for the answer**
> >
> > Thank you for the detailed explanation. You are correct that reconstruction guidance requires the Tweedie estimate of the samples, whereas your method does not. I now realize I overlooked this distinction earlier, and your clarification has resolved most of my concerns.
> >
> > Regarding the additional baseline, I appreciate the authors' efforts in providing new results. However, I noticed that the performance of C-PGDM is significantly worse compared to the results reported by the original authors, and its flow-based variant (C-PGFM) has not been included. That said, I understand that reporting C-PGFM can be challenging due to its complex hyperparameter tuning, so I will not insist on its inclusion.
> >
> > Lastly, while the additional perceptual quality metrics (FID, KID) reported by the authors are not ideal, I believe this is a minor issue in the context of the overall contribution, which I still acknowledge and value. I have increased my score accordingly.

---

> > > ### Author Response · Authors · 2024-11-22
> > >
> > > We thank Reviewer V1uU for the timely reply and the acknowledgement of our contributions. Regarding the C-$\Pi$GDM baseline, we want to point out that it works well on noiseless measurements. We carried out experiments with NFE = 5 and 0 measurement noise on one thousand images from the CelebA_HQ dataset. The tasks are 4x super resolution, Gaussian deblurring, and 90% random inpainting. The results are as follows:
> > > | C-$\Pi$GDM | PSNR | SSIM | LPIPS | FID |
> > > |------------------|-----------------|-----------------|-----------------|-----------------|
> > > | Super Resolution | 29.72 | 0.8312 | 0.0790 | 28.44 |
> > > | Deblurring | 28.19 | 0.7894 | 0.1020 | 28.73 |
> > > | Inpainting | 24.40 | 0.6787 | 0.1896 | 47.93 |
> > >
> > > We see that under noiseless measurements, i.e. $\sigma_n = 0$, C-$\Pi$GDM performs well on 4x super-resolution and Gaussian deblurring, especially for the LPIPS metric. Our results in the noiseless case are comparable to that of Table 4 in the C-$\Pi$GDM paper for super-resolution and deblurring, although they experimented on FFHQ. For the noisy tasks, we tune the hyper-parameters carefully, but still fail to obtain good results on both Gaussian deblurring and 90% random inpainting, which might suggest that C-$\Pi$GDM struggles to deal with noisy case. This suggests an advantage of FIG.

---

### Official Review · Reviewer_BWWP · 2024-11-03

**Soundness:** 3
**Presentation:** 2
**Contribution:** 3
**Rating:** 6
**Confidence:** 4

**Summary:**

The authors propose a method, FIG, to pre-train a flow matching or diffusion model to be used as a prior to solve linear inverse problems. FIG considers the conditioning on the measurements in the diffusion process. However, instead of using the direct measurement y, it conditions on the “measurement interpolants” y_t, as it generates x_t. The authors show that for specific choices of the interplant, the relationship between the signal x_t and the measurement y_t hold with the same linear model of the inverse problem (and adjusted additive noise magnitude).
Theoretical guarantees are provided for the correctness of the proposed optimiser and sampler. FIG is also tested on several inverse problems (SR x4/x16, Gaussian/Motion deblurring, inpainting) and yields state of the art results.

**Strengths:**

The idea of measurements interpolants seems intuitively good and is clearly supported by the experiments (with state of the art performance especially in very challenging cases).
The authors also introduce the background in both flow matching and diffusion models, and provide some theoretical proof that the proposed method is valid given that Assumption 1 holds.
The proposed method in essence is quite simple and easy to implement (as a change to existing uses of pre-trained flow matching or diffusion models).

**Weaknesses:**

The presentation could be improved to explain much more clearly the motivation behind FIG.
What are the issues that other methods could not address and that FIG does address?
My current understanding is that one _can_ condition on the measurement interpolants and then in the conclusions there is a comment that it is easier to create the interpolants than to denoise the signal.
I would suggest to include a specific subsection that explains:
1) What limitations exist in how other methods condition on measurements
2) How measurement interpolants address those limitations
3) Why conditioning on interpolants is theoretically or empirically advantageous

It would then be necessary to demonstrate in the experiments how these limitations are not addressed
by other methods but are addressed by FIG.

**Questions:**

1) I would like to know the motivation for the measurement interpolants. I have an intuition of why they are used, but I could not find that in the paper and I would like the authors to explain; More precisely, I am asking what was wrong with the conditioning of other methods and how this new approach makes that right.
2) I would like to have an explanation for what each of the theorems contributes. Possibly this is connected to the previous point. I suggest to 1) add a brief paragraph after each theorem explaining its practical implications; 2) include a discussion subsection that ties together how the theorems collectively justify or enable the proposed method; 3) provide intuitive explanations or examples to complement the formal mathematical statements.

---

> ### Author Response · Authors · 2024-11-21
>
> We thank reviewer BWWP for the valuable feedback. We address your concerns and questions below.
>
>  ***QA1: What limitations exist in how other methods condition on measurements?***
>
> > We find methods of high reconstruction quality are often time consuming, and efficient methods fail to consistently obtain high quality reconstruction throughout different tasks. Furthermore, none of the existing methods are able to handle challenging scenarios like high measurement noise. Thus we see FIG, an algorithm that could efficiently handle both easy and hard image restoration tasks, as a significant improvement for linear image restoration with diffusion/flow matching prior. Please refer to the updated Section 2.3 for more detailed discussions on the limitations of existing methods.
>
> ***QA2: How measurement interpolants address those limitations? Why is conditioning on interpolants theoretically or empirically advantageous?***
>
> > By incorporating measurement interpolants defined in Eq. (13), FIG’s conditional updating scheme only involves simple gradient descent on an L2 loss of a linear function of the signal (Algorithm 1 step 7-9), which substantially improves the efficiency. The experiments show that the simple updating scheme of FIG is not only effective for low noise tasks, but robust to high noise as well.
>
> > Theoretically, we show in Eq. (20) that the updating scheme is a numerical discretization of a probability flow ODE, whose corresponding dynamics is shown in Theorem 1, justifying the consistency of our algorithm. Corollary 1 demonstrated that the FIG updating direction is equivalent to maximizing a regularized log-likelihood, providing further intuition of the algorithm.
>
> ***QA3: I would like to know the motivation for the measurement interpolants. I have an intuition of why they are used, but I could not find that in the paper and I would like the authors to explain; More precisely, I am asking what was wrong with the conditioning of other methods and how this new approach makes that right.***
>
> > The intuitive idea of measurement interpolants comes from filtering [1], such that the measurement $y$ is interpolated to mimic an observation process {$y_t$}$_{t\in[0,T]}$. However, traditional filtering methods such as Sequential Monte Carlo are extremely slow, and corresponding empirical performances fail to match SOTA baselines. In view of the drawbacks, we consider using gradient descent instead.
> We also want to point out that the conditioning of other methods is not wrong, they are just different ways of incorporating the measurements into the prior model. See the last paragraph in Section 2.3 for a comprehensive review of existing methods. Comparatively, FIG’s way of incorporating measurement information is empirically shown to be efficient and robust to measurement noise, demonstrating an advantage on existing benchmarks.
>
> ***QA4: I would like to have an explanation for what each of the theorems contributes. Possibly this is connected to the previous point. I suggest to 1) add a brief paragraph after each theorem explaining its practical implications; 2) include a discussion subsection that ties together how the theorems collectively justify or enable the proposed method; 3) provide intuitive explanations or examples to complement the formal mathematical statements.***
>
> > Thanks again for the suggestions. We’ve updated the paper accordingly in Section 3.3.
>
> Reference:
> [1] Dou, Zehao, and Yang Song. "Diffusion posterior sampling for linear inverse problem solving: A filtering perspective." The Twelfth International Conference on Learning Representations. 2024.

---

> > ### Comment · Reviewer_BWWP · 2024-11-25
> >
> > Dear authors, thanks for your answers.

---

> ### Author Response · Authors · 2024-11-26
>
> Thanks a lot for your time and valuable feedback. We hope that our response and the updated paper answered your questions. Please let us know if you have any further questions or comments. Thank you once again, and we look forward to your response.

---

### Author Response · Authors · 2024-11-21
**General Response**

We thank all reviewers for their time and feedback. We are excited to see that our work was assessed as well-written by all reviewers; “intuitively good and clearly supported by the experiments”, “quite simple and easy to implement” by reviewer BWWP; “well-written with both theoretical claim and experiment results, can be deployed in both diffusion and flow matching scenarios” by Reviewer V1uU; “distinguishes itself from existing methods, presents a well-defined algorithm with strong theoretical justification and comprehensive empirical evaluations”, “offers a promising approach for tackling linear inverse problems”, and “suggesting a significant contribution to the field” by Reviewer 9aCd; “theoretical framework is well-written with mathematical formulations” by Reviewer iHSi.

We updated the paper and appendix in blue color to address reviewer questions. Additions include: 1)  more detailed explanations of the algorithm motivation and some intuitions on the analysis of the algorithm in Subsection 2.3 and 3.3; 2) a most recent baseline [1] in Section 4 and Appendix E; 3) a comparison on FID and KID in Appendix D, together with a detailed discussion on why we do not think they are appropriate evaluation metrics;  4) a more detailed ablation study on the effectiveness concerning the variance rescaling parameter of measurement interpolants in Appendix C.2; 5) a discussion on potential extensions to bilinear problems in Section 5.

We answer questions for each reviewer individually and point to the corresponding paper and appendix additions.

[1] Pandey, Kushagra, Ruihan Yang, and Stephan Mandt. “Fast Samplers for Inverse Problems in Iterative Refinement Models.” In NeurIPS (2024).

---

### Comment · Reviewer_9aCd · 2024-12-03
**Thanks for the authors' responses and efforts**

Thanks for the authors' responses and efforts. I keep my rating.

---

### Meta-Review · Area_Chair_iFM2 · 2024-12-22

**Metareview:**

Summary.

The authors propose a method, FIG, to pre-train a flow matching or diffusion model to be used as a prior to solve linear inverse problems. FIG considers the conditioning on the measurements in the diffusion process. However, instead of using the direct measurement y, it conditions on the “measurement interpolants” y_t, as it generates x_t. The authors show that for specific choices of the interplant, the relationship between the signal x_t and the measurement y_t hold with the same linear model of the inverse problem (and adjusted additive noise magnitude). Theoretical guarantees are provided for the correctness of the proposed optimiser and sampler. FIG is also tested on several inverse problems (SR x4/x16, Gaussian/Motion deblurring, inpainting) and yields state of the art results.

Strengths.

The idea of measurements interpolants seems intuitively good and is clearly supported by the experiments.
The paper is well written with a good background discussion, experiments, and theoretical analysis.

Weaknesses.
The motivation for measurement interpolants can be improved.
A minor weakness/limitation is that the proposed method is not directly applicable to non-linear inverse problems or models with latent encodings.

Missing.
The paper is missing comparisons with some existing methods.

Reasons.
The paper proposed an interesting and efficient approach for solving inverse problems. Overall, strengths of the paper outweigh the weaknesses.

**Additional Comments On Reviewer Discussion:**

The paper had good discussion among reviewers and authors.

Reviewers mainly asked for clarifications and comparison.
Authors provided detailed responses and several concerns were addressed.

Overall, all reviewers are leaning toward accept.

---

### Decision · Program_Chairs · 2025-01-22

Accept (Poster)